# Ant collective cognition allows for efficient navigation through disordered environments

Aviram Gelblum[1], Ehud Fonio[1], Yoav Rodeh[1,2], Amos Korman[3]*, Ofer Feinerman[1]*

[1]Department of Physics of Complex Systems, Weizmann Institute of Science, Rehovot, Israel; [2]Department of Software Engineering, Ort Braude College, Karmiel, Israel; [3]The Research Institute on the Foundations of Computer Science (IRIF), CNRS and University of Paris, Paris, France

**Abstract** The cognitive abilities of biological organisms only make sense in the context of their environment. Here, we study longhorn crazy ant collective navigation skills within the context of a semi-natural, randomized environment. Mapping this biological setting into the 'Ant-in-a-Labyrinth' framework which studies physical transport through disordered media allows us to formulate precise links between the statistics of environmental challenges and the ants' collective navigation abilities. We show that, in this environment, the ants use their numbers to collectively extend their sensing range. Although this extension is moderate, it nevertheless allows for extremely fast traversal times that overshadow known physical solutions to the 'Ant-in-a-Labyrinth' problem. To explain this large payoff, we use percolation theory and prove that whenever the labyrinth is solvable, a logarithmically small sensing range suffices for extreme speedup. Overall, our work demonstrates the potential advantages of group living and collective cognition in increasing a species' habitable range.

*For correspondence:
amos.korman@irif.fr (AK);
ofer.feinerman@weizmann.ac.il
(OF)

## Introduction

Movement and navigation are key ingredients in the ecology of any animal species (*Nathan et al., 2008*). Within its environment, an animal may encounter diverse and unpredictable navigational challenges. In some cases, such as chemotaxis, a simple biased random walk strategy suffices for efficient navigation (*Berg, 2000*). However, when challenges are complex (*Vergassola et al., 2007*), the animal may need to exploit cognitive tools (*Lihoreau et al., 2019*) such as active sensing of the environment (*Gomez-Marin et al., 2011*), processing of gathered information (*Vergassola et al., 2007*), and memory formation (*Collett et al., 2013*). Indeed, an animal's navigation strategies reflect both the structure and statistics of its environment (*Dyer, 1998*) and its cognitive capacities (*Geva-Sagiv et al., 2015*; *Collett et al., 1998*).

Cooperation is a common means by which animals may increase their cognitive capacity (*Couzin, 2009*). Group living animals may improve their navigational choices through social learning (*Mueller et al., 2013*), collective decision making (*Couzin et al., 2011*; *Simons, 2004*), and leadership (*Gelblum et al., 2015*). Whether these forms of collective cognition enable a species to broaden the range of navigational challenges it can overcome (*Couzin, 2009*) is an intriguing question.

We approach this question within the context of cooperative transport (*Czaczkes and Ratnieks, 2013*) by longhorn crazy ants (*Paratrechina longicornis*) (*Feinerman et al., 2018*). To capture the structure and diversity of natural environmental conditions, we track groups of ants as they cooperatively transport large objects through semi-natural environments which mimic random stone-riddled

terrains. The inherent randomness of this setting produces a wide distribution of navigational challenges that facilitates a study of the connections between individual capabilities, environmental statistics, and emergent collective cognition (*Gordon, 2019*).

An additional advantage of considering disordered environments is that motion through such environments has been extensively studied from a physics and mathematical perspective (*Isichenko, 1992*). Namely, percolation theory studies the structure of porous or disordered media by modeling them as discrete or continuous (*Feng et al., 1987*) randomly connected networks (*Stauffer and Aharony, 2018*). The percolation threshold of a network specifies the degree of connectivity at which it undergoes a phase transition. Below the threshold, connections are few and the system breaks into small disconnected clusters. Above the threshold, there are enough connections to form a single giant component which spans the entire system. The 'Ant-in-a-Labyrinth' framework (*Stauffer and Aharony, 2018*; *de Gennes, 2009*; *Feng et al., 1987*; *Straley, 1980*; *Hughes, 1995*; *Berger et al., 2003*; *Kozma and Nachmias, 2009*; *Ben Arous et al., 2016*; *Richardson et al., 2011*) studies physical flows through porous media by considering the motion of a biased random walker as it traverses a percolation network. Importantly, while in these physical settings the dynamics are memoryless and governed by purely local forces, biological systems are not necessarily limited by these constraints; animal navigation employs memory (*Collett et al., 1992*) and may include non-local strategies such as collective sensing (*Berdahl et al., 2013*) or pheromone trails (*Reid et al., 2012*). The 'Ant-in-a-Labyrinth' framework therefore allows for an interesting comparison between the performances of passive physical systems and cognitive biological systems.

## Results

### Ants-in-a-Labyrinth

Semi-natural labyrinths were created by randomly spreading uniform sized cubes (with a footprint of 0.8 by 0.8 $cm^2$) across a planar arena (70 by 50 $cm^2$) bounded from three directions and open toward the nest (see *Figure 1*). The ants were initially recruited into the maze arena using cat food morsels, until a clear trail was established to the initial load location near the center of the board's edge that is furthest from the entrance (see *Figure 1b*). The cat food morsels were then removed and instead a large food-like item (1 cm radius silicon ring) was placed in this initial location (See Materials and methods). This artificial load was made attractive to the ants by storing it overnight in a closed bag of cat food (*Gelblum et al., 2015*). The ants were then allowed to carry the food without any external intervention. Each maze configuration was tested once, before repeating the process of maze creation, recruitment, and carrying.

In order to deliver the load to the nest, the ants had to cooperatively transport it amid cubes which often interconnect into composite obstacles (see *Video 1*). These obstacles generally interfere with the motion of the large load but are effectively transparent to individual ants that can easily pass in the small gaps between adjacent cubes (*Fonio et al., 2016*; *Figure 1a*). This discrepancy makes escaping local traps and consequently finding a winding trajectory that crosses the labyrinth highly non-trivial (*Figure 1*, *Figure 2a*).

The entire carrying process was filmed and the coordinates of the load, ants, and cubes extracted using image processing (see Materials and methods, *Source datas 1–2*).

A labyrinth was declared to be solved if the load reached the edge of the arena closest to the nest within an 8-min time frame. By comparison, in the absence of cubes, the load traverses the same distance in a mean time of less than 1.5 min. In the language of percolation theory, higher cube coverage (see *Figure 1b*, Appendix 1.1, *Figure 1—figure supplement 1a*) corresponds to reduced connectivity between the regions that are available to the load's motion. Low and intermediate cube densities that correspond to a connectivity level above the percolation threshold yield soluble mazes. As cube density grows, the intricacy of the maze rises; this manifests in a reduction in connectivity of the allowed regions, as the percolation threshold is approached. At a certain high enough cube coverage, the labyrinth falls below its percolation threshold. This is accompanied by the formation of large composite obstacles that break the labyrinth into disconnected islands which render it insoluble (*Figure 2b*). We find that the performance of the ants decreases with the number of cubes comprising the maze: sparse mazes were more likely to be solved, were crossed faster, and with a shorter trajectory arc length (*Figure 2b–c*, *Appendix 2—figure 2b*). The ants were able to

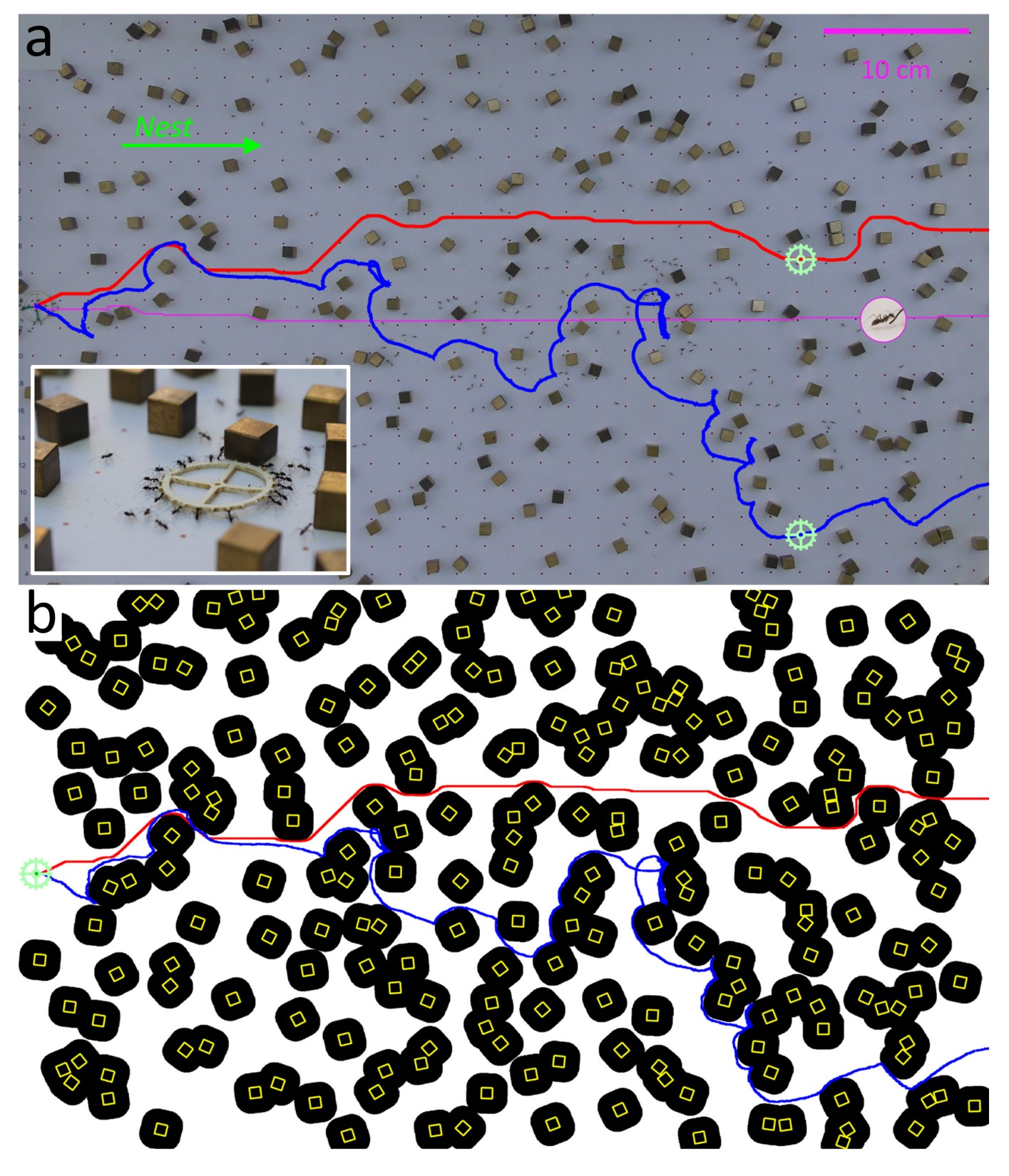

**Figure 1.** Motion within a maze. (a) Setup for cube maze experiments. Overlaid are the measured load trajectory (blue), shortest possible path for the load (red) and shortest possible path for ants (magenta). Inset shows a close-up image of the ring-shaped load as it is carried by ants through the cube

*Figure 1 continued on next page*

*Figure 1 continued*

maze. A sample video of the ants' motion is provided in *Video 1*. (**b**) Cube coverage of the maze shown in (**a**). Black regions are areas that are inaccessible to the load's center, taking into account its radius. Cube coverage is defined as the fraction of inaccessible areas (Appendix 1.1, *Figure 1—figure supplement 1a*). The load is marked in pale green and shown at its initial location. Shortest available path for the load is plotted in red and the ants' actual trajectory is drawn in blue, as in (**a**).

The online version of this article includes the following figure supplement(s) for figure 1:

**Figure supplement 1.** Fraction of forbidden space and dense maze solving probabilities.

---

solve mazes up to cube coverage of 55% (300 cubes). This number is not far from the percolation threshold of this system, which occurs at 60% coverage, and beyond which there is a sharp decrease in the number of solvable mazes (see Appendix 1.2, *Figure 1—figure supplement 1b*).

## Ants outperform biased random walks

To evaluate the ants' performance under the percolation threshold, we compared it to simpler, non-biological models of motion in which the ants' attraction to the nest is mapped to a directional bias. Specifically, we introduce the *pinball model* as a continuous version of the discrete biased random walk. This model describes the viscous motion of a ring that falls through an array of square pegs (*Halperin et al., 1985*) in the presence of Brownian noise (see Materials and methods). Notably, the pinball model significantly outperforms the discrete biased random walk (see Materials and methods, *Appendix 2—figure 2c*). This improved performance stems from the fact that, unlike the biased random walk which can stall at any obstacle, the falling ring quickly bypasses isolated pegs by rolling over them. Similar rolling behavior is also evident in the ants' collective motion (Appendix 1.3 and *Appendix 1—figure 1*; *Czaczkes and Ratnieks, 2011*).

The free parameters of the pinball model were fit so that its simulated trajectories (see *Geva-Sagiv et al., 2015*) reproduce major features of the ants' collective motion in the absence of cubes (see Materials and methods). Fixing these parameters, the simulation was then run over all cube configurations as extracted from the experimental footage (200 instantiations per cube configuration, see trajectory heat map example in *Figure 2a*). As expected, increased cube coverage renders the simulation less effective in terms of success probability, solution times and total trajectory arc length (*Figure 2b–c*, *Appendix 2—figure 2b*).

We go on to compare the performance of the pinball model to that exhibited by the ants (*Figure 2b–d*). By construction, in the absence of cubes the pinball model performs similarly to the ants. This similarity carries over to low-density mazes, which were mostly composed of isolated cubes, since both the ants and the pinball simulation quickly roll across these small obstacles. At intermediate cube densities, where composite obstacles are present, the ants outperform the physical model by a gap that widens with increasing cube number. Finally, both algorithms are similarly ineffective at solving very dense mazes. The ants' performance surpasses not only that of the pinball model but also variants of this model with other noise statistics (see *Figure 2d* - local, non-responsive algorithms, blue points/axis, Appendix 2.1, 2.2 and *Appendix 2—figures 1*, *2*). *Figure 2d* summarizes the comparisons between empirical ant performances and those of different numerical simulations and is referred to below as further models are introduced.

## Collective extension of sensing range

Percolation mazes can be viewed as a collection of disjoint traps (*Berger et al., 2003*; *Figure 3b*). Therefore, to identify the crucial ingredients which help the ants outperform local physical models we focused on motion within such traps. Much like local maxima in

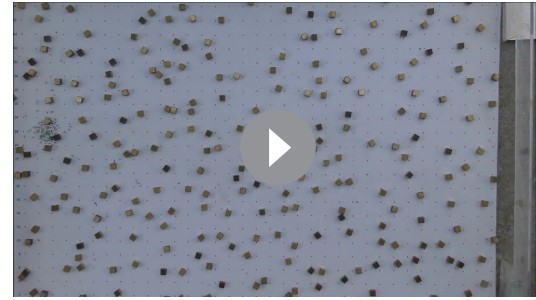

**Video 1.** An example of cooperative transport of a 1 cm radius ring-shaped load across a 260 cubes maze. The nest is located to the right. The video is sped up X8 of real-life speed.
https://elifesciences.org/articles/55195#video1

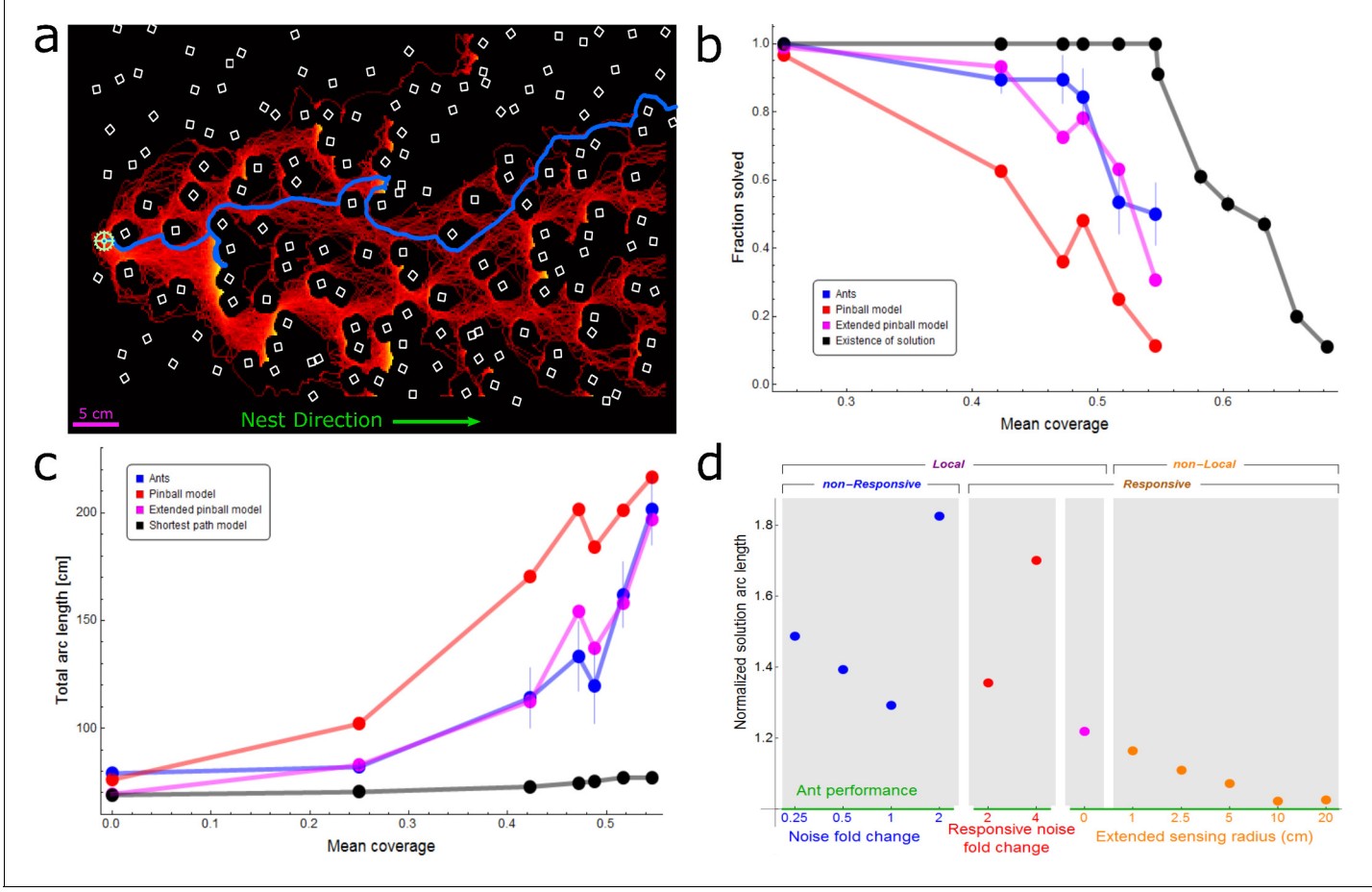

**Figure 2.** Ant vs simulation performances. (**a**) Heat map of trajectories of 200 simulation iterations over an example maze (brighter colors signify more visits, cubes are drawn in white). Actual ant trajectory for this maze is overlaid in blue. Initial location for all trajectories is marked by a green cogwheel. (**b**) Probabilities to solve the maze as a function of mean coverage, for ants (blue), pinball model (red), and extended pinball model (magenta) simulations. The percent of solvable mazes is depicted in black (up to 0.55 coverage - experimental mazes, 0.55 coverage and above - computer generated mazes). Sample sizes (from small coverage to large): Ants - 15,57,19,19,28,30, Pinball Model - 200 iterations each over 10,14,10,8,15,11 distinct mazes, Extended Pinball Model - 500 iterations each over 10,14,10,8,15,11 distinct mazes. Existence of Solution - (experimental - up to 0.55 coverage): 10, 14, 10, 8, 15, 11 (generated- 0.55 coverage and beyond): 100 for each coverage. (**c**) Comparison of average total arc length of ants' and different types of simulations' trajectories (color scheme as in (**b**)). The geodesic shortest path traversing the maze is shown in black. We take into account the different success rates of the simulation and ants as shown in panel (**b**) by adding a penalty to each iteration/experiment which was not successful. The added penalty equals average speed multiplied by the time stuck before termination of experiment/iteration. Error margins in (**b,c**) are standard errors of the mean. Wherever no error is visible, the error is small enough to fit within the filled circle marker. Sample sizes (from small coverage to large): Ants - 31,10,14,10,8,15,11, Simulations - as in (**b**) except the first point is 200/500 iterations in the no cubes case, Shortest Path - 10,14,10,8,15,11, first point is simply the width of the board. (**d**) The performance of different simulated models normalized by empirical ant performance (marked by horizontal green line). We use a single inverse measure for the performance of the simulations, $\frac{L_{\mathrm{sim}}}{L_{\mathrm{ants}}}$, where $L$ is the solution arc length (calculated as in panel (**c**)) averaged over all cube densities. Models are categorized by their locality and responsiveness, and separated into three differently colored x-axes; each corresponding to a different kind of simulation, wherein the numeric value is the main parameter we change in that simulation. Local non-responsive models are versions of the pinball model where noise levels were varied (Blue dots over blue axis, a noise value of 1 is the fitted value in original model. Appendix 2.1 and *Appendix 2—figure 1*). Local responsive models are versions of the pinball model in which noise is temporarily altered in response to the load being stuck in a trap (Red dots over red axis, Appendix 2.3 and *Appendix 2—figure 3*) or a new random bias direction is temporarily selected (Magenta dot over orange axis, Appendix 2.2 and *Appendix 2—figure 2*). The non-local responsive models are versions of the extended pinball model with different sensing radii (Orange dots over orange axis, Materials and methods, Appendix 2.4, *Appendix 2—figure 4*). For a full version of this panel with three additional simulations with considerably inferior performance, see *Appendix 2—figure 5*.

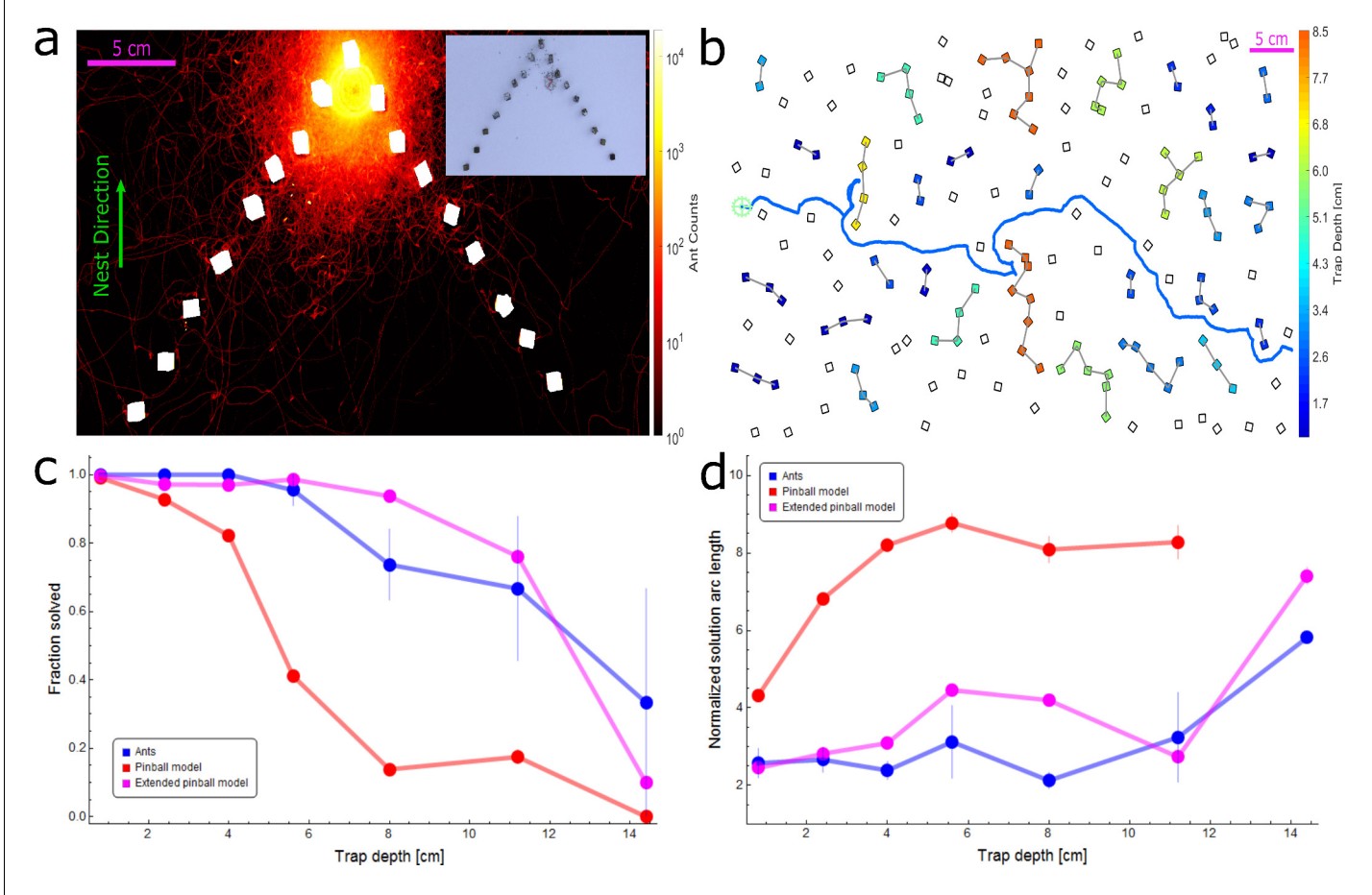

**Figure 3.** Simulation and ant performance near traps. (a) Logarithmic heat map showing the spread of ants while the load is located near the deepest point of a triangular trap (extracted from 23 min of footage). The nest direction is towards the top. Color intensity represents the total number of ant counts within each 2D bin over the aforementioned experimental duration. A $r^{ants}_{sense} \cong 10$ radius area centered on the load contains ~99% of ant traffic in the vicinity of the load (see *Figure 3—figure supplement 1*). Inset shows an example image from the video footage of the experiment. (b) Illustration of traps on a sample maze. Each group of cubes comprising a trap are connected by gray lines and colored according to the trap depth in cm (as defined in Appendix 1.5) corresponding to the color bar. The empirical ant trajectory for this particular realization is plotted in blue (initial location marked using a pale green cogwheel). Nest direction is to the right. (C) Probability of trap solution as a function of trap depth for ants (blue), pinball model (red), and extended pinball model (magenta). Sample sizes (from shallow trap to deep): Ants - 73,70,35,22,19,6,3, Pinball Model - 2645,2886,1646,1289,982,343,105, Extended Pinball Model - 8979,8203,4637,3395,2042,815,403. (d) Average normalized arc length of the trajectory taken to solve a trap as a function of trap depth for ants and simulations (color scheme as in (c)). Trajectory lengths are normalized by trap depth (see Appendix 1.5, Materials and methods). Ant performance is approximately constant up to $D = 12$ cm which is on the scale of $r^{ants}_{sense}$ (see panel (a)). Sample sizes: Ants - 73,70,35,21,14,4,1, Pinball Model - 2620,2675,1352,530,136,60,0, Extended Pinball Model - 8952,7969,4497,3347,1913,620,302. Error margins in (c,d) are standard errors of the mean. Wherever no error is visible, the error is small enough to fit within the filled circle marker.

The online version of this article includes the following figure supplement(s) for figure 3:

**Figure supplement 1.** Cumulative ant spread.
**Figure supplement 2.** Trap depth distributions.

optimization problems, traps are areas in which motion toward the global solution is blocked. Escape from a trap must therefore be facilitated by secondary forces that are not aligned with the general bias. Similar to common optimization heuristics (*Kirkpatrick et al., 1983*), in the pinball model these forces are the result of random noise. The ants, however, exhibit more elaborate motion. We find that when the carrying group enters a trap, its characteristics of motion change; specifically, they spend a higher percentage of the time walking against the bias (Appendix 1.4, *Appendix 1—figure 2*).

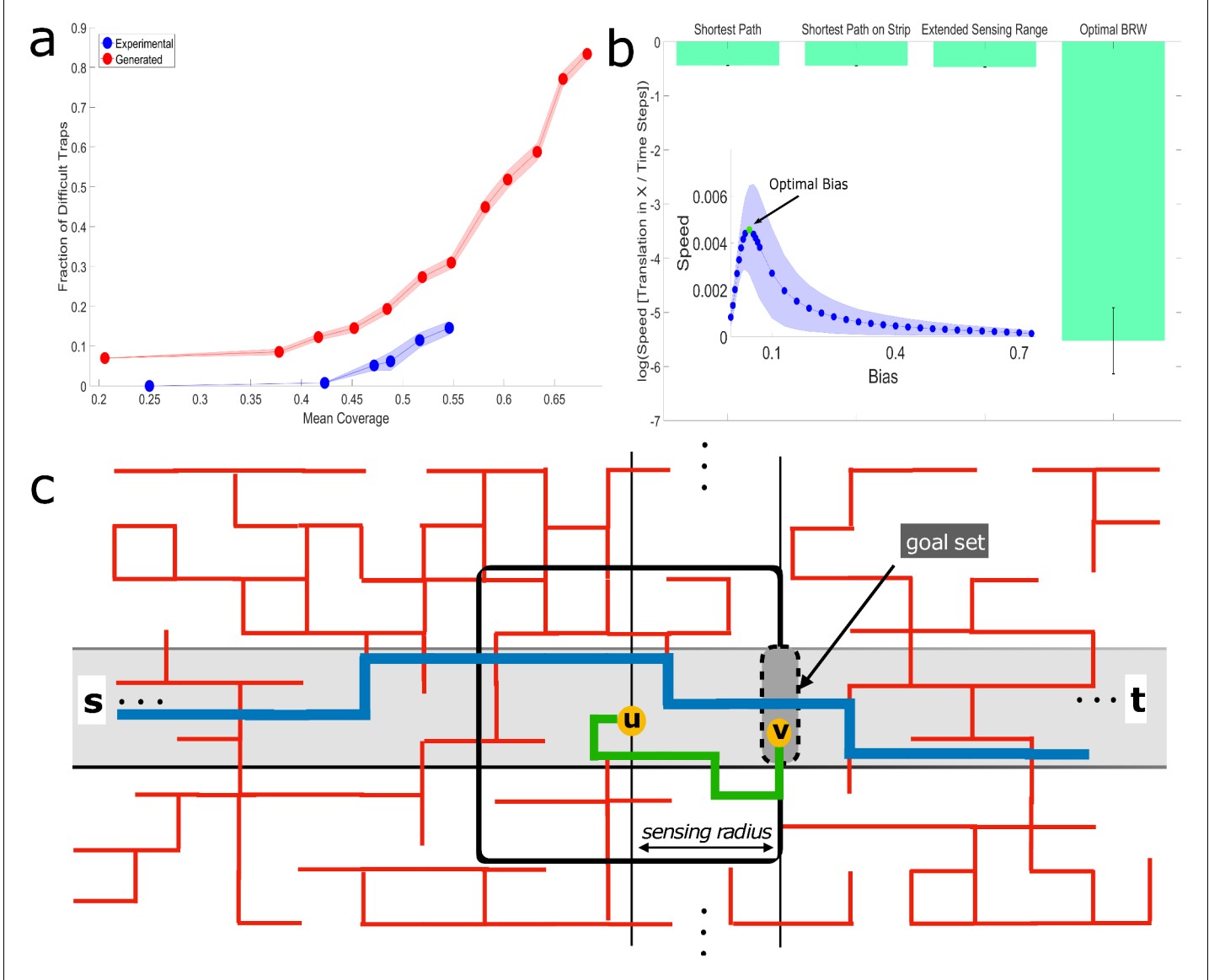

**Figure 4.** Efficiency of logarithmic range extended sensing. (**a**) The fraction of cubes which belong to difficult traps, out of the total number of cubes in the system, as a function of mean coverage of the cube maze. Difficult traps are those defined by $D>r_{sense}^{ants}$. Note the sharp increasing trend above 0.55% coverage. Error margins are standard errors of the mean. Sample Sizes (from small coverage to large): Experimental (number of cubes in the calculation) - 1017,2511,2033,1631,3380,2798, Generated - 50 different mazes for each cube number: 100,200,225,250,275,300,325,350,400,425,450. (**b**) Simulated performances of percolation lattice solution algorithms just above the percolation threshold ($p = 0.55$). The biased random walk model whose bias, $B = 0.045$, is optimized (**Berger et al., 2003**) to increase drift speed (see inset). Note that this a local-sensing algorithm with this optimized bias stillperforms significantly worse than a logarithmic extended sensing algorithm. The extended sensing algorithm is only slightly worse than the overall shortest path and the shortest path that is constrained within a logarithmic width strip crossing the maze. Error bars in the main panel and shaded regions in the inset signify standard deviation. Sample sizes: Main figure - calculation for the first 3 bars is one number per maze. The last bar is a simulation with 200 iterations over each maze. Since we used 50 different lattice configurations, the sample size is 50,50,50,10000. Inset - 200 iterations over 50 different lattices; thus, 10,000 samples per point. (**c**) Schematic illustration of the theoretical extended-sensing algorithm on a 2D percolation grid (see Materials and methods, Appendix 3.1, 3.2). Red lines are the open edges of the infinite cluster across which the walker moves from an initial point s to a final point t. The walker moves by executing a series of short bouts. Depicted in the image is a single bout wherein the agent, currently positioned at point u, accesses information within its sensing range (black square, of logarithmic radius) and advances along the green geodesic (fully contained within the sensing range) to some point v on the next goal set line. Such bouts allow the agent to cross the maze on a path whose distance is extremely close to that of the shortest path (blue line) between the initial point *s* and the final point t, that is contained within a strip of logarithmic width (colored gray).

The online version of this article includes the following figure supplement(s) for figure 4:

**Figure supplement 1.** Carried loads stay within a confined strip.

It was previously shown for ants (*Ron et al., 2018*; *Gelblum et al., 2016*) (and other animal groups *Tunstrøm et al., 2013*; *Buhl et al., 2006*) that physical interaction with a trap can induce change in the collective characteristics of motion. This responsiveness does not require any individual to be explicitly aware of the trap and can therefore be perceived as implicit, emergent trap detection. However, our simulations show that mere responsiveness to local information does not suffice in explaining the ants' enhanced performance (see local responsive algorithms in *Figure 2d*, Appendix 2.2, 2.3, *Appendix 2—figures 2c*, *3*).

Beyond the effect of local mechanical collisions, the collective motion of *P. longicornis* is known to be guided by information that is brought in by newly attached transient leader ants (*Gelblum et al., 2015*; *Gelblum et al., 2016*). Once attached, these ants steer the entire group and determine the collective direction of motion. Leader ants come from the non-carrying population which surrounds the load (*Gelblum et al., 2015*; *Fonio et al., 2016*). Their attachment therefore allows carrying ants to use information that is beyond the load's immediate locality and could enable the group to collectively extend their sensing range (*Berdahl et al., 2013*). Next, we estimate the distance at which information is gathered and assess the impact of this form of non-locality on global performances.

To approximate the sensing radius, we focused on the spatial distribution of non-carrying ants around a trapped load (*Figure 3a* and Materials and methods). We find that when the load is delayed within a trap, non-carrying ants spread across a circular region whose outer radius, $r_{sense}^{ants}$, is on the order of 10 cm (*Figure 3a*, *Figure 3—figure supplement 1*). Although a relatively small fraction of the ants reach areas that are $r_{sense}^{ants}$ centimeters away from the load, this is the relevant length scale to consider; this is since even a single leader ant suffices to steer the entire group and guide it as far as 10 cm (*Gelblum et al., 2015*). Hence, when the load is delayed within an obstacle, leader ants constantly present the carrying group with potential crossing routes up to a 10 cm radius. Collectively, this implies that a number of potential routes are presented in parallel to the carrying group. In turn, the coordinated motion allows the group to explore the suggested traversal routes (*Gelblum et al., 2016*) until, eventually, they find an escape route that bypasses the obstacle (*Fonio et al., 2016*). Indeed, we find that preventing individual ants from entering the trap from detour routes significantly reduced the extent of the ants' collective exploration within the trap (see Appendix 1.7 and *Appendix 1—figure 4*).

## Extended sensing facilitates efficient trap and labyrinth traversal

To assess the contribution of the extended sensing to trap negotiation, we considered an *extended-pinball model, an extension of the pinball model* with an enlarged sensing range, $r_{sense}$ (see Materials and methods). This is a responsive model in which obstacle sensing induces temporary change in the direction of the bias. Unlike the responsive local models described above (*Figure 2d*), in the extended pinball model the choice of the temporary directional bias is affected by non-local environmental structure. Specifically, the direction of this temporary bias was chosen to lead toward a point along the obstacle's boundary that is conducive to bypassing the obstacle, entails minimal directional changes (*Gelblum et al., 2015*; *Forster et al., 2014*), and is no further than a distance of $r_{sense}$ from the load's center (for more details see Materials and methods). We ran computer simulations of this model over the experimentally acquired cube maze configurations - 500 instantiations per cube configuration.

Next, we compared the effectiveness of trap escape by the ants, the pinball model and the extended pinball model. To do so, we defined the depth of a trap as the length of the geodesic required to escape its deepest point (Appendix 1.5 and *Appendix 1—figure 3*). We then quantified how well the ants and the simulations perform when facing traps of a given depth independent of the overall complexity of the maze. This was done by assessing the average distance travelled to escape the trap and normalizing it by trap depth. In the basic pinball model, this ratio increases with trap size as would be expected from a random walker that relies on rare large fluctuations to escape. The ants do much better: up to trap depths that roughly coincide with the measured upper bound on their sensation range, $r_{sense}^{ants}$, the ants' escape route is highly efficient, namely it scales linearly with trap depth (see *Fonio et al., 2016*). For traps that are deeper than $r_{sense}^{ants}$ the ratio quickly rises. The extended pinball model highlights the role that sensing range plays in trap escape. To efficiently bypass a trap of a given size, the sensing range must be at least as large (see *Appendix 2—figure*

*4c*). Specifically, setting the sensing range parameter of the extended pinball model to its experimentally measured upper bound $r_{sense} = r_{sense}^{ants}$ yields trap solution performance similar to that of the ants (*Figure 3c-d*).

We now turn to check how non-local information and the resulting improvement in negotiating medium-sized traps (i.e. up to $r_{sense}^{ants}$) reflect on overall performance. We find that the extended pinball model simulations with $r_{sense} = r_{sense}^{ants}$ performed significantly better than the original pinball model and almost matched the performance of the ants (see *Figure 2b–c*). In addition, we found that simulating the extended pinball model with values of $r_{sense}$ that are smaller than $r_{sense}^{ants}$ diminished performance. Conversely, increasing the value of $r_{sense}$ beyond $r_{sense}^{ants}$ had no effect on overall performance (see *Figure 2d* - orange points/axis, Appendix 2.4 and *Appendix 2—figure 4a,b*).

We note that while the performances of the extended pinball model with a sensing radius of $r_{sense}^{ants}$ are comparable to those of the ants, they are still slightly inferior (*Figure 2b–c*). This may be expected due to the relative simplicity of this model which does not aim to precisely replicate the distributed nature and navigational capabilities of ants. Rather, this model is intended to capture the ants' extended sensing range and demonstrate the navigational importance of collecting information beyond the physical boundaries of the load.

The relation between the ability to escape a single disjoint trap and overall performance in crossing the entire terrain relies on the statistics of trap sizes in the environment. Indeed, we find that below the ants' solution threshold of 55% coverage, close to the system's actual percolation threshold, the vast majority (93.6%) of the traps are smaller than $r_{sense}^{ants}$ (*Figure 4a*, Appendix 1.6, *Figure 3—figure supplement 2*). The ants' efficient performance at the global level can therefore be traced to their ability to quickly overcome traps up to this size. Moreover, the rarity of large traps renders larger sensing ranges unnecessary. Next, we present theoretical analysis to make these intuitive points more precise.

## Logarithmic sensing radius suffices to approximate the shortest path

Percolation theory deals with statistics of cluster sizes on random graphs while the Ant-in-a-Labyrinth literature examines motion over such graphs. These fields of study could therefore provide firm theoretical grounds for studying the relations between environmental statistics and collective navigation as found in our experiments.

A main result of the ant-in-a-labyrinth literature is that a pure random walker would cross the percolation maze in a time that scales quadratically with the size of the system (*Ben Arous et al., 2016*). Moreover, adding a small bias to the random walk results in much faster passage times that are linear in system size (*Berger et al., 2003*; *Reichhardt and Reichhardt, 2018*). Further increasing the bias does not necessarily increase speed since the walker tends to get trapped. This implies the existence of an intermediate bias in which traversal speed is maximized (*Fribergh and Hammond, 2014*; *Bénichou et al., 2014*; *Barma and Dhar, 1983*) - we verified this theoretical result by simulating random walks with different biases on percolated square lattices (*Figure 4b*). In all these cases, the sensing range of the walker is, by definition, zero. It is therefore interesting to compare these performances to those of an ant-inspired random walker with an extended sensing range.

Our main theoretical result concerns the impact of moderately extending the sensing range (*Angel et al., 2008*) to be logarithmic in system size. We first used simulations to show that such a modest extension can lead to a huge (over 200-fold) speed up in traversal times when compared to classical ant-in-a-labyrinth solutions (*Figure 4b*, Appendix 3.2). Then, to better understand the origin of this result, we combined mathematical analysis and simulation (*Figure 4c*) to show that a walker whose sensing range is logarithmic in system size can cross the labyrinth along a path that approximates the shortest possible path to extremely high precision (Appendix 3.1, 3.2, Materials and methods, *Appendix 3—figure 4a*).

We next present an outline of the formal arguments of our proof which are laid in detail in Appendix 3. Our analysis can be broken into three parts: First, we prove that two distant points on a percolation grid above the percolation threshold (p=0.5) can be connected along a path that is fully confined to a narrow strip (*Figure 4c*). Second, we use numerical calculations to show that the length of this confined path is extremely close to the length of the shortest possible path between these two points. Finally, we provide an algorithm for a mobile agent with a logarithmic sensing range

which allows the agent to proceed along a path that is extremely close to the confined path and, therefore, to the shortest possible path between the two points.

More specifically, we considered a percolated grid above the percolation threshold, and two points $s$ and $t$ that belong to the infinite connected component. For the first aforementioned part, we aim to prove that with high probability there exists a path that connects these two points and is completely contained within a strip $S$ of logarithmic width around the aerial line that connects the two points (colored gray in *Figure 4c*). Essentially, this result follows from a result by *Aizenman and Newman, 1984* which states that, above the percolation threshold, the probability of obtaining a given obstacle decreases exponentially with its size. This implies that if the aerial distance between the points is d, then with high probability, there will not be an obstacle larger than $W = c \log d$, for some constant c > 0, which blocks the aerial line between them. Taking the width of the strip $S$ to be slightly larger than $W$ ensures that, with high probability, there is a path which is contained in $S$ and bypasses these obstacles. Having established the existence of such a path, we denote the length of the shortest of all such paths by $\tilde{D}$.

Next, we numerically demonstrated that $\tilde{D}$ is extremely close to $D$, the unrestricted shortest path between s and t (*Figure 4b*). This was done by first generating a random lattice slightly above the percolation threshold (p=0.55). We then defined a narrow strip that traverses the lattice and calculated the shortest path from side to side, where the path is either unconstrained and can include any vertex on the entire lattice (length $D$) or constrained to stay bounded to the strip (length $\tilde{D}$). These shortest paths were calculated by finding the regional minimum of the summation of two geodesic distance transforms over image representations of the random lattice, with the two edges of the strips acting as seed locations. To find the shortest path constrained to the strip, we simply ran the same calculation on the subset of the maze which only contains the strip. We find that the average percent of increase to the length of the shortest path when constrained to the aforementioned strip is merely ~0.46% (averaged over N=50 lattices of size d = 70,000,).

Finding a path whose length approximates $\tilde{D}$ may not be a trivial task for an agent with a small vision-radius. As our main theoretical result, we prove that a logarithmic field of view, $r = b \log d$, suffices to yield paths that closely approximate the length of $\tilde{D}$. In fact, by appropriately choosing the constant $b$ we can guarantee that the length of the resulting path will approximate $\tilde{D}$ to any desired approximation. To achieve this, the agent executes a series of short bouts where each allows it to reduce its aerial distance to the destination, $t$, by roughly $\log d$ (*Figure 4c*). At the beginning of a bout the agent assesses all paths that start at its current location (node $u$ in *Figure 4c*), are contained within its sensing range, $r$ (black square in *Figure 4c*), and lead to some point $v$ in the strip $S$ (colored gray in *Figure 4c*) which is roughly $\log d$ closer to the destination (node $v$ in the 'goal set' in *Figure 4c*). It then advances along the shortest of these paths (which exists with high probability). Since the bout starts and ends in $S$, any deviation from $S$ stays within the radius $r$, and is hence small (*Figure 4c*). Since the sensing radius, $r$, is larger than the width of the strip, the trajectory chosen by the agent can be shown to be extremely close to the shortest path that is fully contained in the strip and advances the same distance. Stringing these bouts allows the agent to cross the maze on a path whose length is extremely close to $\tilde{D}$ and, in turn, to the shortest possible distance $D$.

## Relating theoretical results and empirical findings

The theorem outlined above shows that a small logarithmic sensing range suffices for fast traversal of a percolation maze. Our theoretical results further indicate that a route that is confined within a narrow strip around the aerial line connecting the start and end points can well-approximate the shortest path possible. In other words, the proof suggests that efficient labyrinth crossings do not require significant deviations from the aerial line. In line with this suggestion, we find that the empirical load trajectories are typically confined to relatively narrow strips, even at high cube densities (*Figure 4—figure supplement 1*).

To further interpret our experimental results in light of our theory, we must first return to our underlying assumptions. While in our experiments we vary the density of open edges $p$, in our theoretical results we assume a fixed value $p_0$ which is above the percolation threshold. To reconcile these analyses, we note that for a sufficiently large system size, $N$, the dominant factor in the sensing range required to solve the maze would be $\log N$. This logarithmic sensing range then suffices for the entire range of mazes with $p \geq p_0$, that is, mazes of the same size whose coverage is lower.

Our theoretical analysis thus predicts a logarithmic relation between system size and sensing range. An algorithm implementing this sensing range can efficiently navigate most solvable mazes of the corresponding size. We next turn to apply this result to quantitatively relate two length scales: the size of the ants' foraging range which, in the case of this species, is on the order of 10 m (*Jaffe, 1993*), and the scale of extended sensing which is on the order of 10 cm. To make this relation, we must specify a third length scale - the spacing of the abstract grid used in our proofs. We note that grid spacing coincides with the length of a cube's edge which is 1 cm. Indeed, the addition of a single cube translates to the removal of an edge in the percolation grid. We further note that both cube size and experimental load radius are not arbitrary. They were both chosen to coincide with the typical size of the loads cooperatively transported by longhorn crazy ants (*Gelblum et al., 2015*; *Feinerman et al., 2018*). Smaller obstacles will not stall the carrying group. Larger, extended obstacles can no longer be approximated by a percolation network.

With these numbers in hand we can now verify whether the ants' natural sensing range is congruent with our theoretical results. Given the 1 cm grid spacing, a foraging range of 10 m coincides with a system size of $N = 1000$. According to our theoretical results, the expected sensing range at this system size is on the order of $\log(1000) \approx 10$. Translating the answer back into centimeters, we find that the ants' sensing range is expected to be on the order of 10 cm. This length scale coincides with our empirical findings regarding both the ants' sensing range and the strip width to which their collective solutions are confined (*Figure 4—figure supplement 1*).

We wish to stress that these measures are not meant to be precise. First, our experimental system's length is 70 cm, which is substantially smaller than the ants' maximal foraging range. This is not a major concern since optimal sensing ranges are robust across system sizes due to their logarithmic nature. The optimal sensing range for a 70 cm system is only $\log(1000)/\log(70) = 1.6$ times smaller than the sensing range that corresponds to a 10 m foraging range, and is still on the order of 10 cm. Second, there is no reason to believe that the ants are optimally tuned for the environments studied in this paper or for a specific system size. We merely claim that the sensing range we measured is extremely efficient for traversing disordered systems of varied sizes and densities. It is this kind of generality one might expect from natural navigational systems that must deal with a large number of unexpected challenges.

## Discussion

An organism's survival depends on its ability to overcome challenges toward reward. The evolution of such abilities can be affected by various factors including the difficulty of the challenge, its prevalence (*Schlaepfer et al., 2002*), the reward it entails (*Krill, 2007*) and the energetic cost of maintaining cognitive and physical capabilities required to tackle it (*Burns et al., 2011*). Accommodating these possibly conflicting considerations can lead to evolutionary trade-offs in problem solving abilities (*Isler, 2013*; *Mendl, 1999*; *MacIver et al., 2010*; *Shoval et al., 2012*). The navigation behavior we describe may be the result of such a trade-off: the ants use their distributed nature to probe the surroundings non-locally but only moderately extend their sensing range. The extreme navigational efficiency induced by this moderate increase in sensing range stems from the fact that it matches the statistics of trap sizes in percolation networks. Indeed, percolative environments, either below or above the percolation threshold, hardly exhibit any traps of intermediate (i.e., super-logarithmic and sub-linear) size (*Stauffer and Aharony, 2018*) and navigational strategies to tackle such traps are thus useless.

The ants use remote, active, collective sensing to probe their surroundings. Remote sensing is extremely common in the biological world (*Klemas, 2013*). Primary examples are the use of sight, olfaction, hearing, and vibration (*Hill, 2001*; *Klärner and Barth, 1982*). Animal remote sensing also extends to the use of more active tactics such as echolocation (*Au, 1997*) and active electrolocation (*Albert and Crampton, 2005*). Most ant species are known to use eyesight to assist their navigation (*Wehner et al., 1996*). However, since ants are physically small in comparison to the smoothness of the surfaces they inhabit, their lines of sight along these surfaces are inevitably short. Thus, sight alone may not suffice to bypass local obstacles during cooperative transport. Instead, the ants use their numbers to actively extend their sensing range by sending out scouts in all directions. Indeed, evolutionary trade-offs as discussed above can be expected to be prevalent in cases of active sensing (*Arditi et al., 2015*).

This brings us to the second aspect of the ants' extended sensing; namely, the fact that it is collective. It is not uncommon that animal groups engage in collective sensing. For example, the 'many eyes principle' describes the ability of a group of prey animals to share surveillance efforts, such that the first to spot an approaching predator can warn the rest (*Treherne and Foster, 1981*). Another striking example comes from fish shoals; golden shiners use collective sensing to track environmental features that are unavailable to individuals and only make sense on the scale of the group (*Berdahl et al., 2013*). This collective effect is reminiscent of the ants' collaborative navigation scheme studied here. Indeed, as a group, the ants manage to find navigational solutions to large obstacles that are imperceptible to any single individual (*Fonio et al., 2016*).

The 'ant-in-a-labyrinth' problem was originally suggested by Pierre De-Gennes as a means of investigating diffusion through disordered media (*de Gennes, 2009*). It applies, for example, to the motion of an electron in a metal-insulator alloy under an electric field and at some finite temperature (*Gefen et al., 1983*; *Stanton et al., 1986*; *Nava et al., 1976*). The electron can be modeled as a random walker on a percolation network where the effect of the electric field is captured by a drift term and the effect of temperature by an additional random component. This biased random walk framework underlies most ant-in-a-labyrinth literature (*Stauffer and Aharony, 2018*; *de Gennes, 2009*; *Feng et al., 1987*; *Straley, 1980*; *Hughes, 1995*; *Berger et al., 2003*; *Kozma and Nachmias, 2009*; *Ben Arous et al., 2016*; *Richardson et al., 2011*). Inspired by the ants' behavior, we took a more algorithmic perspective to this problem. Instead of studying the properties of a walker with a given set of local rules fixed by the laws of physics, we explored the impact of extending the sensing range on navigational performances. Such studies regarding the effects of locality on performances are, in fact, a common theme in theoretical computer science (*Peleg, 2000*). In general, local algorithms are often preferred for their simplicity. However, it is known that they can fall short under different circumstances (*Peleg, 2000*; *Linial, 1992*; *Naor and Stockmeyer, 1995*; *Goos et al., 2017*; *Sarma et al., 2012*). Indeed, we have seen that in our system the performance of physics-based local algorithms is substantially inferior to the ants' performance. Conversely, extending the sensing range to be logarithmic in the size of the grid can have a significant impact on navigation time, overshadowing purely local solutions (*Kirkpatrick et al., 1983*; *Fonio et al., 2016*; *Deneubourg et al., 1983*).

Finally, the wide applicability of percolation theory leads us to hypothesize that similar relations between environmental structure and perception range may carry over to other biological systems. These include populations that occupy an extended area in either physical (*Berdahl et al., 2013*; *Reid and Latty, 2016*; *Nakagaki, 2000*) or abstract (*Wagner, 2005*) space. Spreading allows the population as a whole to sample the space in a non-local manner. As an example, robustness and neutral mutations allow an evolving population to spread over areas in fitness space. This non-locality enables parallel sampling of the fitness landscape and increases the ability of the population to incorporate advantageous mutations (*Wagner, 2005*).

## Materials and methods

### Experimental setup: percolation experiment

Data was collected from two nests of *Paratrechina longicornis* in the Weizmann Institute of Science area, Rehovot, Israel. Tests were carried out during the summer when these ants display collective transport behavior (*Trager, 1984*). Experiments were conducted on a $70 \times 50$ cm board on which ants were allowed to cooperatively carry heavy loads. In each nest site, the testing board was positioned according to the availability of appropriate filming conditions (flat floor and a sufficiently large area with uniform illumination). As *P. longicornis* are a polydomous species, a 3-sided plastic frame was place around the board, with the opening directed towards the largest nest entrance. This was done to make sure the bias the ants exhibit is directed towards the same nest direction, i.e. there are no conflicting biases.

Before each experiment, a specific amount of cubes were randomly spread over the board. Ants were then recruited using Royal Canin cat food. The cat food morsels were gently picked up and moved backwards several times until a clear trail was established to the initial load location near $(x, y) = (0, 25)$ on the board. The cat food morsels were then removed and instead the ants were given an artificial ring-shaped 1.5 mm thick, 1 cm radius silicon load. The artificial objects were

stored in advance overnight in a closed bag of cat food from the same brand, to make them attractive to the ants. The board and load were marked with different colors to facilitate image analysis and tracking.

After recruitment and positioning of the load at the initial location, the carrying process through the cube maze was allowed to unfold without intervention. The entire process was recorded using a Panasonic HC-VX870 camcorder at a 4K resolution with a frame rate of 25 frames per second in most cases (a small fraction of the experiments were recorded at HD resolution with a frame rate of 50 frames per second).

Experiments were declared to be over if one of three conditions was fulfilled:

1. The ants were able to solve the maze; that is, the load exited the board through the edge close to the nest.
2. After a minimum of 8 min of experiment, if the ants were not able to solve the maze.
3. The ants were able to overcome the cubes by climbing over them with the load. As this behavior was displayed only when the load was very much stuck, these experiments were considered as unsuccessful trials (i.e. - the ants were considered unable to solve the maze).

Each maze was tested once, before repeating the process of maze creation, recruitment and carrying.

## Experimental setup: wedge experiment

Unsolvable wedge-trap experiments were performed to assess the spatial distribution of non-carrying ants around the load while it is trapped. These experiments were conducted on a single colony within the Weizmann Institute of Science, Rehovot, Israel. Here, the board was a blank A3 page which was put within a dedicated elevated perspex arena open on one side, with a paper ramp connected to it. The open side was directed towards the nest entrance. Two different set-ups were tested: a wedge-shaped unsolvable trap was created either by manually setting cubes ~1.5–2 cm apart (a composite trap), or by appropriately positioning two perspex plates (a single entrance trap). Only the entrance in the latter set of experiments was also composed of cubes, to produce the same difficulty in the front of the trap. The ants were recruited using a procedure similar to the one used for the percolation experiment (see above section), and then allowed to carry the load for extended periods of time (i.e. hours). These experiments were recorded using a Panasonic HC-VX870 camcorder at HD resolution with a frame rate of 50 frames per second.

## Image processing

Videos were analyzed using custom code built in MATLAB. One program was dedicated to tracking the motion of the center as well as the orientation of the load, based on iterative HSV thresholding of the image to recognize the colored markings on the load. Ants carrying the load were also recognized by transforming the image into grayscale and performing homomorphic filtering before applying a threshold. Ant blobs were distinguished from other blobs based on features such as circularity and eccentricity.

Cube locations were recognized by another specialized program, through a combination of HSV and RGB thresholding. Cube blobs were automatically recognized and subsequently manually corrected using a GUI. Cube base locations were then extrapolated from the obtained cube blobs.

The original video had a small effect of pincushion distortion which was accounted for using a spatial distortion fixing transform. Load trajectories and cube locations were corrected.

## Calculation of trajectory arc length of single trap solutions

In *Figure 3d*, we show the mean arc length obtained for crossing single traps of different depths. To calculate this value, we considered the relevant trajectory section to begin when the ant team/simulation reaches a point 1 cm away from a trap, and ends when it advances 3.2 cm ahead in the nest direction (positive x direction), thus assuring the trap is solved. This distance is in line with the distance used for trap definition (Appendix 1.5). The extra 3.2 cm are then deducted from the arc length. The arc length is then normalized by the trap size, $D$.

## Simulations

### Physical simulations

All physical simulations were written based on CapSim (*Tassa, 2019*), a MATLAB based physics engine aimed at simulating multiple 2D rigid body mechanics. Based on our experimentally extracted cube locations, we used CapSim to define the cubes and the edges of the board as collidable immovable objects. The load was defined to be a disk of radius $R = 1.1$ cm, based on the experimental load size ($R = 1$ cm). The addition of 0.1 cm is a result of evaluating simulation results allowing the load to pass through gaps the ants could not. This correction compensates for inaccuracies in cube recognition due to image processing errors and difficulty in assessing manually the cubes' exact location due to their angle relative to the camera. At $R = 1.1$ cm there was a strong correspondence between the ants' and the simulated load's ability to pass through gaps.

CapSim allows manipulation of gravity $g$ (analogous to the bias towards the nest), drag $\mu$, and object mass m. We also defined a random noise force term $\nu$ which is recalculated every time step and added to the gravity term. The force direction is sampled from a uniform distribution, and its size is sampled from a normal distribution with mean 0 and standard deviation $\sigma_F$. This parameter is important to simulate the inherent noise of the biological system in question.

After fitting model parameters (see relevant section below), the simulation was run over all experimentally implemented mazes (200/500 iterations each), allowing the dynamics to unfold up to a maximum time of $T_{max}$.

### Discrete biased random walk over continuous cube mazes

This simulation implements discrete biased random walk of a disc of radius $R = 1.1$ cm, moving across the continuous cube mazes extracted from the experimental footage. The simulation was written in MATLAB. The walker moves over the continuous board with a discrete step of size $S = 0.1$ cm. The direction of motion is randomly assigned in every time step, where the probability of going towards the nest (to the right) is biased such that $p_{right} = 0.25 + B$ and the other three directions are equally likely $p_{left} = p_{up} = p_{down} = 0.25 - \frac{B}{3}$, where $B$ is the bias parameter. At every time step, the simulation checks if the load's suggested motion direction leads to overlap with any of the cubes. If so, the direction is re-selected randomly; otherwise, the step is taken in the selected direction. The edges of the board are treated as impassable walls.

After fitting model parameters (see relevant section below), the simulation was run over all experimentally implemented mazes (100 iterations each), up to a maximum duration given by $T_{max}$.

### Simulations on discrete lattices

This set of simulations was developed to complement our mathematical proof regarding the efficiency of the vision algorithm compared to biased random walk, on a dense percolation maze. To do so, we created random percolation lattices poised just above the percolation threshold (which is 0.5 for bond percolation on the $\mathbb{Z}^2$ lattice), $p = 0.55$. In line with the theoretical proof (Appendix 3.1), in these simulations, $p$ is the probability of an edge to be open or accessible. In all the simulations described in this section, the walker moves over the giant component induced by the open edges of the lattice. 50 random lattices of dimensions $NX\delta \log_2(N) = 70000X120 \log_2(70000)$ were generated. Following the theoretical considerations described in Appendix 3.1, a concentric strip of width $\alpha \log_2(N) = 20 \log_2(70000)$ (1/6 of the width of the lattice) was defined as the 'internal strip'.

All simulations start at a node which is included in the giant component, closest to the center of the leftmost column of the aforementioned internal strip. The goal of the simulations is to traverse the maze over the giant component from this initial point to any point on the rightmost column of the internal strip.

As described in the main text, we ran two types of simulations. First, a simple biased random walker simulation was run over all random lattice instances (50 iterations each), for different bias $B$ values, where the bias is defined as in the previous biased random walk simulations (see above). The second is an extended vision algorithm. In this algorithm, the walker has a vision radius of $\gamma \log_2(N) = 20 \log_2(70000)$. Note that the vision radius is equal to the width of the internal strip. At every time step of the simulation, the walker goes along the shortest path within a square of edge size $2\gamma \log_2(N)$, centered around its current location, ending at any point which is both included in

the giant component and contained within the column of the internal strip which is located $\gamma \log_2(N)$ further in the positive $x$ direction, measured from the current location (see *Appendix 3—figure 3*).

We also calculated for each lattice the overall shortest path (denoted $D$) and the shortest path fully contained within the internal strip (denoted $\tilde{D}$), from the leftmost column of the internal strip to its rightmost column.

## Fitting model parameters

### Physical simulations

Our system only has three free parameters since the drag term can be simply set to a constant and incorporated into the other parameters of the system. We therefore set μ to a constant.

The other three free parameters were fit to global features of freely moving collective transport (i.e., no obstacles) - mean trajectory arc length, mean velocity and two parameters describing the velocity-velocity cosine correlation function. The parameter space was searched by running 30 iterations of the simulation without cubes using 10 different values for each free parameter, totaling in 30,000 iterations. The global features yielded by the simulation were then subtracted from the experimental values and normalized to account for the different scales of the parameter values. The parameters of the simulation yielding minimum error were then recognized. This process was repeated three times, shrinking the searched parameter space to the distance between two points of the prior computation.

The fitted values for the original simulation parameters are: μ = 10, g = -5.05, $\sigma_F = 1277.8$, m = 14.8571. The simulation time step is $\Delta t = 0.04$ seconds.

The low persistent noise variation of the simulation uses the following parameter values instead: $\sigma_F = 250$, $\Delta t = 0.4$ seconds.

The simulation maximum duration $T_{max} = 8$ minutes is equal to the experimental maximum allowed duration.

### Discrete biased random walk over continuous cube mazes

This simulation has two relevant parameters. The first - step size $S$, was taken to be 0.1 cm. The value of the step size needed to be small enough to allow motion within traps and be compatible with the scale of the cubes and the entire board. It also needs to be large enough to make the simulations fast enough, and allow the simulation some chance to escape complex traps in reasonable time. We therefore took $S = 0.1$ cm to be of the order of magnitude of the velocity of the ants.

The second parameter, the bias $B$, was fitted using global features of the motion of a freely carried load, in a process similar to that described in the prior section. Here we used the mean deviation in the $y$-direction and the mean trajectory arc length as the global features to fit. The obtained fitted value for the bias for our simulation is $B = 0.2211$.

The simulation maximum duration $T_{max}$ is derived from the average velocity of the ants along the trajectory and the experimental maximum allowed duration. The result of the calculation was multiplied by five to give the simulation greater chances of successfully navigating the cube mazes. The resulting value was $T_{max} = 7200$ time steps.

### Simulations on discrete lattices

We wanted to simulate the algorithm with the minimal vision radius such that the next destination column would be fully visible from any point on the current column, thus $\alpha = \gamma$. We also wanted to compare $D$ with $\tilde{D}$ in a non-trivial way and be able to increase the vision radius if needed, so $\delta > \alpha$ and $\delta > \gamma$ was chosen to accommodate computation power considerations. The maximum time allowed for the biased random walk simulation was 150,000 time steps. The maximum advancement in x for all biases after this running duration made us realize there is no point in running the simulation until the maze is solved, and it is better to use a speed measure obtained from the terminated walks.

## Extended pinball model

The extended pinball simulations are the same as the original simulation except the addition of a module responsible for alerting when the load is trapped, based on total motion in the x-direction in the last few seconds. If the load moved less than $\Delta x_{min}$ in this period of time $T_{compare}$, the load is

considered to be stuck. When the load changes its state from 'free' to 'stuck', it acquires a new bias direction based on the local trap structure (the algorithm calculating these directions is described below). Bias magnitude is constant and always set to the parameter fitted to the ant behavior as explained above. The load then continues its motion in this altered state for a duration $T_{\text{changed}}$, after which it changes its state to 'free', the bias vector reverts to its original direction and it cannot become stuck again for another duration given by $T_{\text{cooldown}}$. This cooldown period is added to make sure that if the load moved backwards it will not immediately switch back into the 'stuck' state.

The parameter values used for all extended pinball models (and temporarily altered noise) simulations are: $\Delta x_{\text{min}} = 0.2$ cm, $T_{\text{compare}} = 3$ seconds, $T_{\text{changed}} = 4.48$ seconds, $T_{\text{cooldown}} = 4$ seconds. The default spatially extended sensing parameter used in the extended pinball simulations is $r_{\text{sense}}^{\text{ants}} = 10$. See Appendix 2.4 for the results of simulations with different $r_{\text{sense}}$ values. The extended pinball model further incorporates time correlated Brownian noise to allow for more persistent motion towards escape. Importantly, correlated Brownian noise alone did not lead to any improvement in global performance (see Appendix 2.2 and *Appendix 2—figure 2*).

The extended pinball simulations depend on the assignment of a new bias direction for the simulation when the load becomes stuck. The assigned gravity direction is pre-calculated based on the local structure of the obstacle hindering the load's advancement. For each maze, we divided the space into $0.5 \times 0.5$ cm squares. We then calculated the bias direction for each square center using the 'dilated cube' maze binary image (see Appendix 1.5) and a spatially extended sensing parameter $r_{\text{sense}}$. The following is a general outline of the algorithm and does omit a few minor details dealing with certain edge cases:

1. Check if the square center falls within a blob. If it does not, continue the calculation using the square center; otherwise:
   a. If the entire square is within the blob, ignore this square and continue to the next one.
   b. If the square contains part of the boundary of the blob, find the point on the boundary closest to the square center. Continue the calculation using this point.
2. Check if there are any blob points in a straight line in the x-direction 0.25 cm in front of the point in question. If not, then the load cannot get stuck in this square and therefore we can ignore this square and continue to the next one.
3. Find the closest trap blob ahead of the point in question.
4. Find the point on the boundary of this trap closest to the point in question. We'll refer to this point as the seed boundary point.
5. Using this boundary point as a seed, calculate the geodesic distance in both directions (top and bottom) over the boundary.
6. Cut two boundary pieces: from the seed boundary point to the point $r_{\text{sense}}$ cm away on the boundary in the top direction. Do the same in the bottom direction.
7. For each boundary piece, find the point with the minimum x-value. We'll refer to these as top and bottom points.
8. Calculate the directions between the seed boundary point and the top and bottom points. Rotate by 15° to make the direction closer to that taken by an ant coming from the back. This is done because the initially calculated directions often cross the trap blobs.
9. Select the new bias direction to be the one closer to the positive x-direction of the two options. This is done to make sure the chosen direction is correct for small traps as well as traps which have an easy solution in one direction. New calculated directions for large traps will point backwards in any case.

## Acknowledgements

We thank Yossi Yovel and Itay Benjamini. This work has received funding from the European Research Council (ERC) under the European Union's Horizon 2020 research and innovation program (grant agreements No. 648032 and 770964). OF is the incumbent of the Henry J Leir Professorial chair. EF is the incumbent of the Tom Beck Research Fellow Chair.

# Additional information

## Funding

| Funder | Grant reference number | Author |
|---|---|---|
| Horizon 2020 Framework Programme | 770964 | Ofer Feinerman |
| Horizon 2020 Framework Programme | 648032 | Amos Korman |

The funders had no role in study design, data collection and interpretation, or the decision to submit the work for publication.

## Author contributions

Aviram Gelblum, Conceptualization, Data curation, Software, Formal analysis, Validation, Investigation, Visualization, Methodology, Writing - original draft, Writing - review and editing; Ehud Fonio, Conceptualization, Formal analysis, Visualization, Methodology, Writing - original draft, Writing - review and editing; Yoav Rodeh, Software, Formal analysis, Investigation; Amos Korman, Conceptualization, Formal analysis, Funding acquisition, Validation, Visualization, Methodology, Writing - original draft, Writing - review and editing; Ofer Feinerman, Conceptualization, Data curation, Formal analysis, Supervision, Funding acquisition, Investigation, Visualization, Methodology, Writing - original draft, Writing - review and editing

## Author ORCIDs

Yoav Rodeh (iD) https://orcid.org/0000-0002-7224-6451
Amos Korman (iD) https://orcid.org/0000-0001-8652-9228
Ofer Feinerman (iD) https://orcid.org/0000-0003-4145-0238

## Decision letter and Author response

Decision letter https://doi.org/10.7554/eLife.55195.sa1
Author response https://doi.org/10.7554/eLife.55195.sa2

# Additional files

## Supplementary files

• Source data 1. Cube locations data set caption: Coordinates of vertices of cube bases, specified in cm, relative to a known (0,0) point marked on the experimental board. Each row in every file corresponds to the four vertices of a single cube, ordered as follows: X1, Y1, X2, Y2, X3, Y3, Y4.

• Source data 2. Load trajectory data set caption: Experimental load trajectories, specified in cm, relative to a known (0,0) point marked on the experimental board. Format is X,Y coordinates as a function of time. Time interval between samples is 0.04 s, except for videos 1440005 and 1440011, where the time interval is 0.02 s.

• Transparent reporting form

## Data availability

Full raw data of both the labyrinths and the ants collective trajectories through these labyrinths were uploaded with this submission.

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

## Appendix 1

# Experimental Results

### 1.1 Cube density and coverage

Different levels of maze difficulty were achieved by spreading different amounts of cubes. However, the number of cubes, though informative, is not a concrete measure for the difficulty of the maze. Thus, we decided to use the mean coverage of the cubes as the measure to use for difficulty. The mean coverage is defined as the fraction of area forbidden to the center of the load. This was calculated using 'dilated cube' mazes as defined in Appendix 1.5. The mean fraction of space excluded from the motion of the load center is given simply by the total amount of 'on' pixels, divided by the total amount of pixels in the image (*Figure 1—figure supplement 1a*). We refer to this measure throughout the article and the supplementary material as 'mean coverage'.

### 1.2 Percolation threshold of cube mazes

While both the ants and the simulations are often not able to solve mazes of 300 cubes (0.55 coverage), the real percolation threshold of the system is higher. Since the mazes are finite, a portion of the mazes will be solvable even at very high densities. However, using computer-generated dense cube mazes we observe a clear trend in solvability probability, where a maze is considered to be solvable if there is a line connecting the allowed segments of a vertical line drawn at $x = maze\ width$ and the closest allowed point to $(0, y_{\text{init}})$ where $y_{\text{init}}$ is half the height of the maze in cm, across a 'dilated cube' maze as defined in Appendix 1.5. At 400–450 cubes (0.65–0.7 coverage), most mazes are unsolvable (see *Figure 1—figure supplement 1b*).

### 1.3 Rolling behavior around small traps

When a load-carrying team of ants encountered a small trap (1–2 cubes), they demonstrated a typical rolling behavior, reminiscent of that of an inanimate round physical object. We calculated the maximum total angle accumulated rotating in one direction in a window of 3 s (=75 frames) starting at the frame of incident upon the trap. We compared the resulting distribution with a control distribution generated by performing the same calculation for non-overlapping stretches of 3 s from the same experiments where the load did not encounter any traps at all. The results are displayed in *Appendix 1—figure 1*. The distributions were found to be statistically significantly different (Kolmogorov-Smirnoff test: $p < 10^{-5}$).

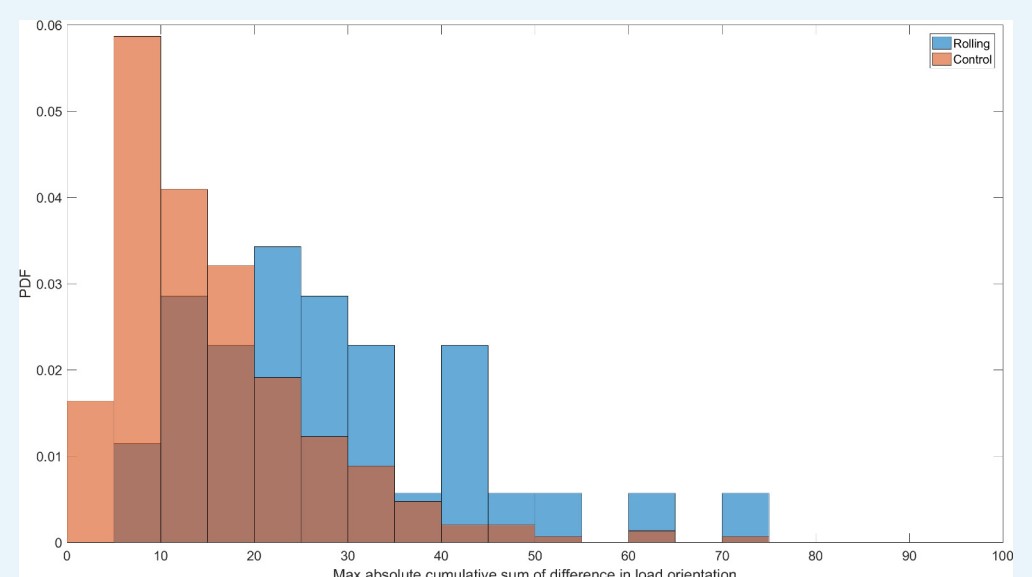

**Appendix 1—figure 1.** Rolling upon impact. Histograms displaying the maximal rotation of the cooperatively carried load in the first 3 s after incident with a small trap (blue) and for stretches of 3 s without any incident (red). The experimental distributions are statistically different.

## 1.4 Comparing the characteristics of trapped backward motion

When trapped in a difficult trap, the carrying ant group's motion characteristics are different from those observed during unhindered, free cooperative transport. Specifically, the percentage of time spent moving backwards in the trapped scenario (28.51%) is >8.5 times larger than in the free motion experiments (3.22%). Similarly, the probability per second to turn backwards is >3 times larger (0.0677 vs. 0.0212, trapped and free motion, respectively).

The ants' motion when trapped also differs from the resulting trajectories obtained in the simple pinball model simulations, also when trapped. Specifically, the maximum distance in each bout of backwards motion, averaged for each trap, is greater in the experimental ant data (1.832 cm) than in the simulation results (1.251 cm, $p<10^{-7}$ Wilcoxon signed-rank test). Importantly, the experimental distribution is far wider than the simulation distribution (0.61 vs. 0.12, experimental and simulation standard deviation, respectively). This width means the ants are more likely to walk backwards further per bout, and thus to solve a difficult trap, see *Appendix 1—figure 2*. Data was limited to the trajectory sections where the load/simulation was stuck in moderate-to-difficult traps ($D>4.8$, see Appendix 1.5 for the definition of trap difficulty). Only traps where data was available for both the ants and the simulations were considered in the calculation. Backward motion bouts were defined by examining the time series of the x-component of the trajectories and searching for regions where the load was further than 0.5 cm away from the deepest point in the trap. Each region was considered a separate bout of backward motion. Thereafter, the maximum value for each region was found.

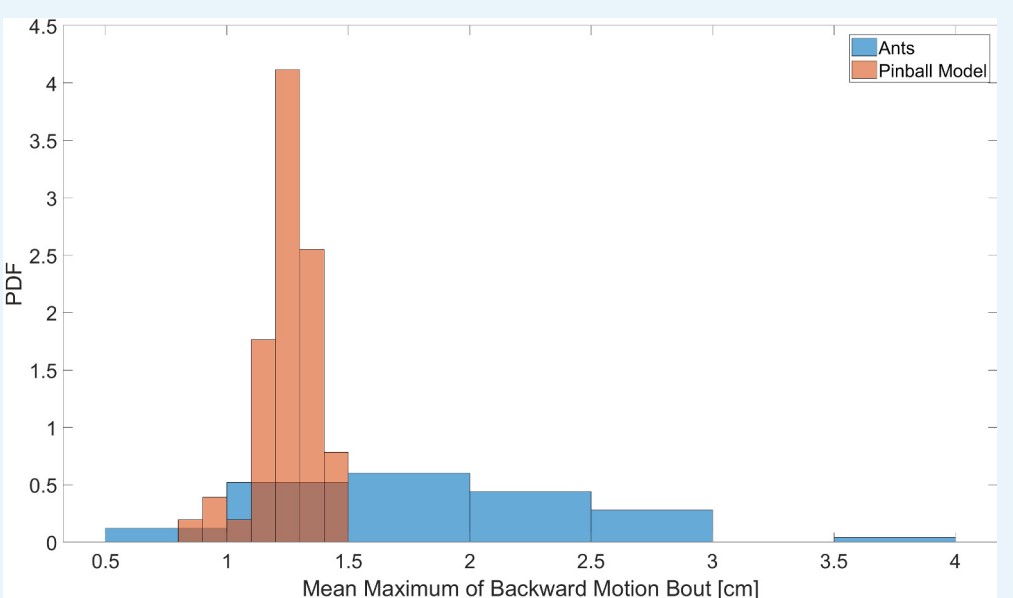

**Appendix 1—figure 2.** Distributions of maximum backward motion. Experimental (light blue) and simple pinball simulation (red) distributions of the maximum point reached during every backward motion bout, that is away from the nest, averaged per trap examined. Note the considerable different width of the distributions.

## 1.5 Trap definition

Individual traps were defined using a geodesic measure. Specifically, we calculated a 'dilated cubes' binary image based on cube locations and radius of the load (=1.1 cm). Namely, we dilated each cube blob by 1.1 cm in all directions. The resulting white regions in the image represent the allowed regions for the load center, and the black the forbidden ones (*Appendix 1—figure 3*). We then sampled points from the load trajectory through the maze at an aerial spacing of 0.5 cm. For each point, we calculated the minimal geodesic path from it to a vertical line drawn 3.2 cm ahead in the x-direction, denoted $L$ (initial point and destination line in green and geodesic path in red and blue (together) in *Appendix 1—figure 3*). The distance forward was approximated from the length of the diagonal of the cube + the diameter of the load, signifying where the trap is most likely solved. This added distance is important to make sure the trap is solved. However, once the ants start traversing this distance, the trap is in fact already solved (see illustration in *Appendix 1—figure 3*). Therefore, the difficulty of the trap is defined to be this calculated minimum geodesic distance ($L$) minus the added 3.2 cm, denoted $D = L - 3.2$ (blue in *Appendix 1—figure 3*).

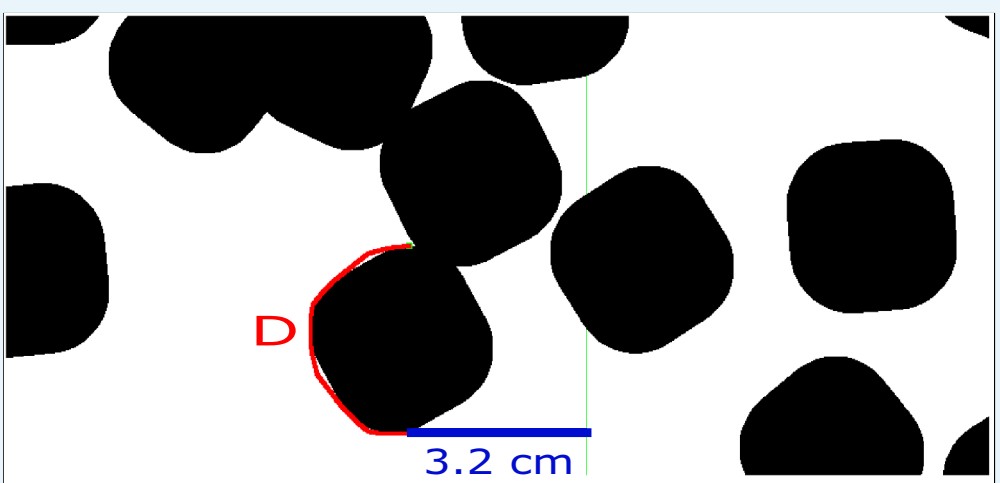

**Appendix 1—figure 3.** Example of Geodesic Measure Calculation. A section of a 'dilated cubes' maze binary image. Each cube blob was dilated by 1.1 cm in all directions to create a map of regions allowed (white) and forbidden (black) for the load center. The small green square is the initial seed and the thin green line 3.2 cm ahead of it is the final destination seed. The red and blue curves together comprise the geodesic path calculated by the algorithm, corresponding to $L$. The red section corresponds to $D$, whereas the blue part is the 3.2 cm extra distance taken to make sure the trap is solved in the calculation. The geodesic distance of this path is used to assess the trap depth after point filtering and clustering as explained in the text of this section.

The points used for the geodesic calculation are a small distance away from each other, to find a good estimation of trap difficulty. However, this means that multiple points may refer to the same trap. In order to cluster points into associated traps, we filtered the trap data by applying a minimum threshold over geodesic distance and then selecting the deepest point in each group of nearby points (using a maximum grouping criterion of 1 cm euclidean distance or 0.5 cm backwards in the x-direction). We used the trajectory time-ordered data to validate and provide an accurate association of points to traps by identifying oscillatory motion patterns which indicate being stuck in a trap.

For *Figure 4a* and *Figure 3—figure supplement 2* we needed to calculate trap depths over entire mazes, including traps the load trajectory did not encounter. To do so, first we identified candidate points using a regional maximum transform over the difference between the x-coordinate of every point relative to the edge of the image (where the experiment ends) and a geodesic distance transform of the dilated cube maze binary image with the seed specified to be the vertical line at x = the image width. We then ran the same geodesic path calculations and trap filtering as described in the previous paragraph, except the grouping distance thresholds used were different (5.5 cm euclidean distance or 4 cm in the x-direction (two-sided)).

## 1.6 Distribution of trap depths

The rarity/prevalence of difficult traps plays a major role in the ability of the ants and simulations to successfully solve mazes of a certain cube density, as implied by *Figure 3*. Thus, we measure trap difficulties over entire experimental and generated mazes. The results are plotted in a bee-swarm type graph in *Figure 3—figure supplement 2*. In line with *Figure 3a*, we see that difficult traps are much more prevalent (18.8%) above 55% coverage, as the system approaches its actual percolation threshold, than under the ants' solution threshold (1.85%). These numbers differ from those displayed in the main text as they disregard the existence of unsolvable traps in the system. This suggests the difference is even greater.

## 1.7 Comparing single-entrance and composite traps

To measure the effect of the ants entering from multiple gaps in a composite trap, we compared the motion characteristics of the load in the two set-ups of the unsolvable wedge experiment; composite, multiple gap cubes trap vs. single-entrance perspex trap (see Materials and methods). The results show a discrepancy in the motion pattern of the load; the load tended to travel further back (Wilcoxon rank sum test, $p = 5.31 \cdot 10^{-5}$) in the composite cube-only trap (*Appendix 1—figure 4*). The maximum distance travelled backwards is calculated for each bout of backwards motion. A bout is considered to begin when the center of the load passes a threshold of $y = 2/3$ cm backwards, relative to the deepest point in the trap (allowed for the load center). This implies the ants entering through the gaps in the back side of the composite trap affect the motion of the load.

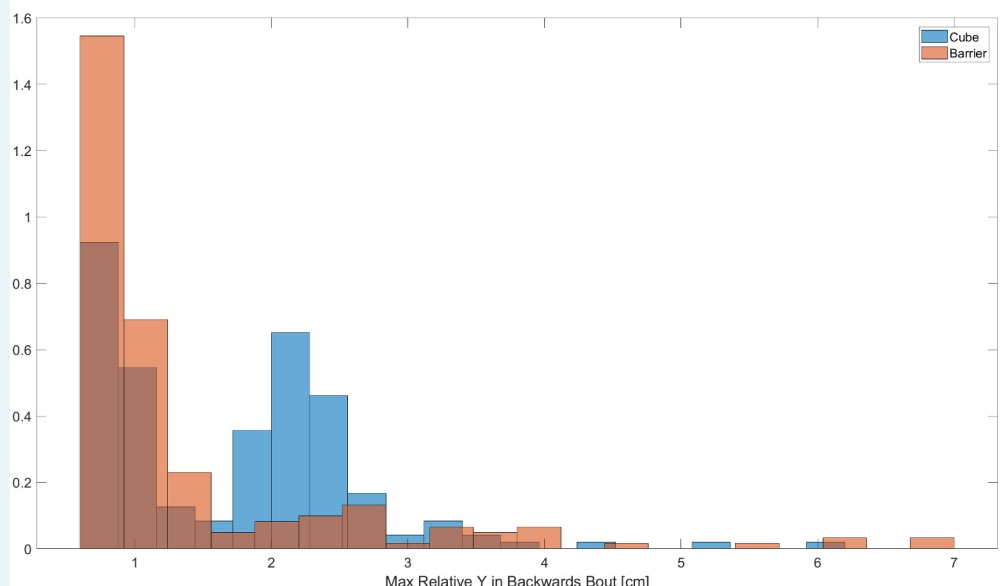

**Appendix 1—figure 4.** Comparison of single-entrance and composite cube traps. Histogram portraying the probability density function of the maximum distance travelled backwards in $y$ in each backward bout for single-entrance (red) and composite (blue) trap setups.

## Appendix 2

# Simulation Results

## 2.1 Cube densities and noise amplitudes in the pinball model

The original simulation used a fitted noise parameter. However, since each cube density yields a different distribution of traps, tuning the noise parameter to cube density might improve the simulation's performance. Namely, at high densities we hypothesized increasing the noise might reduce solution time since the load faces hard obstacles frequently. Conversely, at low densities decreasing noise should strengthen the effect of the bias and allow faster completion with shorter arc length, since large traps are rare. We therefore ran the original simulation using different noise parameter values to assess how its performance compares with that of the ants.

Interestingly, changing the noise does not improve simulation performance (*Appendix 2— figure 1*). In terms of arc length, the original simulation outperforms the other simulations at nearly all densities, with two exceptions. First, in terms of arc length, the X0.5 simulation performs as well as the ants at 0.25 coverage, which is to be expected. It's worth noting that the X0.25 simulation performs worse. This is because with such a low noise value, even the smallest traps poise a problem to the simulation. Second, the X0.25 and X0.5 simulations at 55% mean coverage match the original simulation's performance. At such a high density, the simulations generally do not perform well and the slightly decreased noise does not have a major effect. Simulations with large noise values naturally tend to increase the arc length as the noise parameter dominates the bias and the load performs a random walk across the maze.

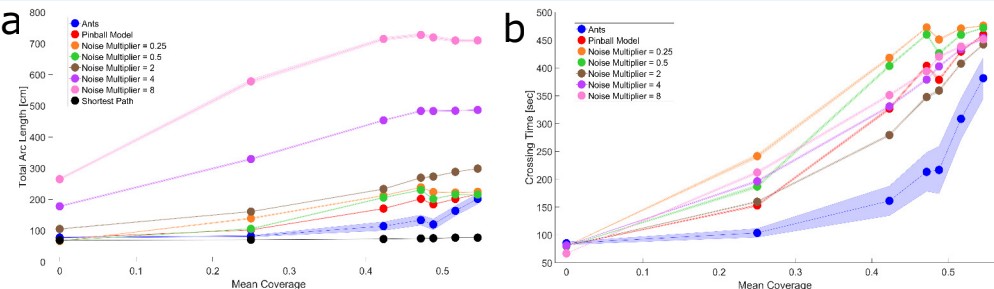

**Appendix 2—figure 1.** Simulations with different noise multiplier values. Plotted are the total arc length (**a**) and solution time (**b**) as a function of mean coverage of traps in the maze for ants (blue) and simulations with different noise parameter fold-change values (as specified in the legends of the figures). The results show that there are no optimal noise parameters per cube density. Generally the original fitted noise parameter performs best for most densities. The ants always outperform the simulations. Shaded regions correspond to standard error of the mean. Wherever no error is visible, the error is small enough to fit within the filled circle marker.

In terms of solution time, the X2 simulation performs similarly to the original at low densities (0.25 mean coverage). At higher densities, the X2 simulation performs slightly better than the original, and the X8 and X4 simulations perform similarly to it. As expected, the high-noise simulations perform better than the low-noise simulations at high densities, since the load can more easily negotiate hard traps with greater noise. However, the improved trap escaping ability comes with the price of inherent randomness, which increases overall solution time, leading to worse performance than simulations with intermediate noise values (original and X2).

It is important to note that the simulations do worse than the ants, at any noise parameter value tested.

## 2.2 Variations of the Pinball Model and Extended Pinball Model

While studying the pinball model and its extension, we varied the noise persistence and size in hopes of getting better results. The rationale behind this change is that lower, persistent noise might help the load escape traps when there's better directional information. Indeed, This change to the noise when combined with the responsive bias scheme of the extended pinball model as explained in Materials and Methods, leads to results close in performance to the ants, as can be seen in *Appendix 2—figure 2*. Thus, what we refer to as 'extended pinball model' in the main text is just the combination of persistent low noise and temporary responsive bias, whose results are represented by the purple lines in *Appendix 2—figure 2*. Note that in and of itself persistent low noise performs worse than the original pinball model. This is not surprising since the noise parameter was fitted to the global features of the ants. However, using the fitted noise values in the responsive bias simulation yields worse results than using the persistent low noise parameter values. This is because the system relies on the local structural information to solve traps, rather than on noisy random walk dynamics. The directional information is important; a simulation with random responsive bias - that is the bias direction temporarily changes when the load is stuck, to a random direction in a 160° arc centered around the negative x-direction - does not perform as well.

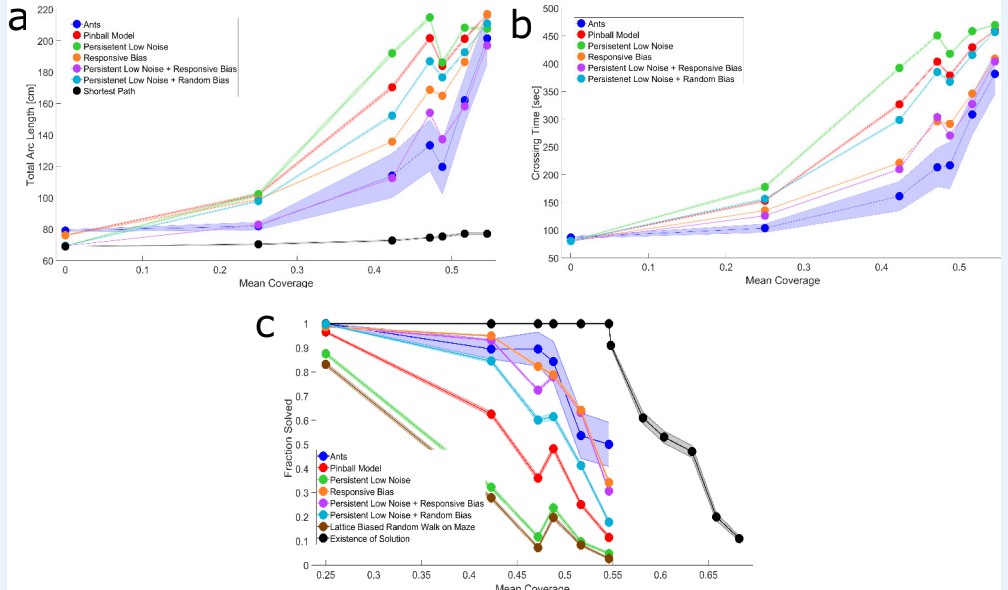

**Appendix 2—figure 2.** Simulation variations. Total arc length (**a**), maze solution time (**b**) and solution probability (**c**) vs. mean coverage for different variations of the pinball model and extended pinball model, combining responsive bias and persistent low noise. The purple line represents the 'extended pinball model' referred to in the main text, and is the best performing simulation of them all. The turquoise line represents a simulation with temporary random responsive bias. Persistent low noise (green) in and of itself performs worse than the original simulation (red), but without it the responsive bias simulation (orange) does not perform as well as with it (purple). The ants outperform all simulations (blue). Black line in (**a**) represents the arc length of the shortest geodesic path across the maze. Black line in (**c**) represents the probability of a maze to be solvable (experimental mazes for coverage ≤ 0.55, computer generated mazes for coverage ≥ 0.55). Brown line in (**c**) represents discrete random walk on a lattice superimposed on the continuous experimental cube mazes. Shaded regions correspond to standard errors of the mean. Wherever no error is visible, the error is small enough to fit within the filled circle marker.

## 2.3 Temporarily altered noise simulations

Instead of temporarily altered bias direction, a possible alternative explanation for the ants' superiority over the simulation regarding trap and maze solution is a temporary increase in noise when the load is stuck in a trap, which would facilitate escape simply by chance. We ran such simulations keeping all other parameters as defined in the relevant Materials and Methods sections, except using the original noise parameters instead of low persistent noise, since we observed that without directional information, low persistent noise simulations tend to perform significantly worse (*Appendix 2—figure 2*). Here when the load becomes stuck, the noise variable - the standard deviation of the force amplitude distribution from which the added random force is sampled - is multiplied by another predetermined parameter $\nu_{\mathrm{mult}}$. The noise reverts to its original value after a certain period of time, similar to the duration scheme defined in the extended pinball model (Materials and methods).

The results (*Appendix 2—figure 3*) show that temporarily altered noise simulations perform terribly in terms of arc length, and slightly better than the original simulations in terms of solution time (but still worse than the extended pinball model and the ants). This is most likely because slightly higher noise simulations still do not efficiently solve traps, and significantly higher noise simulations solve traps efficiently, but often move at high velocities in wrong directions after escaping the trap. This results in erratic, very high arc length trajectories.

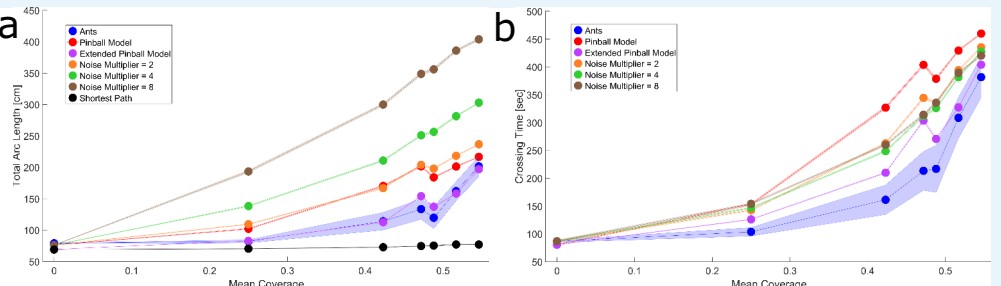

**Appendix 2—figure 3.** temporarily altered noise simulations. Total arc length (**a**) and maze solution time (**b**) vs. mean coverage for simulations implementing an algorithm with temporarily altered noise when the load gets stuck within a trap, for different fold changes of the original simulation noise parameter value. These simulations perform much worse than the original simulation (red) in terms of arc length but better in terms of solution time. However, the ants (blue) and the extended pinball model (purple) perform better than these simulations in both measures. Shaded regions correspond to standard errors of the mean. Wherever no error is visible, the error is small enough to fit within the filled circle marker.

## 2.4 Sensing parameter variation simulations

We estimated the extent by which the ants spatially extend their collective sensing experimentally as described in Materials and Methods and used the obtained value as a parameter (denoted $r_{sense}^{ants}$) when calculating the bias direction the simulated load assumes when stuck in a trap (see Materials and methods). We varied $r_{sense}$ to assess how it affects simulation performance.

In *Appendix 2—figure 4* we observe that for $r_{sense}$ values ($r_{sense} = 1, 2.5, 5$) smaller than the value used in the simulation based on the controlled unsolvable trap experiments ($r_{sense}^{ants} = 10$), the simulations perform worse. The simulation with $r_{sense} = 20$ performs similarly to the $r_{sense}^{ants} = 10$ simulation. The rarity of very large traps (see *Figure 4a*, *Figure 3—figure supplement 2*) means the added information value for such traps is marginal. Moreover, mechanically, the simulation is not statistically likely to walk backwards very far since the change in gravity

direction is temporary. Importantly, the ants always perform better than the simulations, across all $r_{sense}$ values tested.

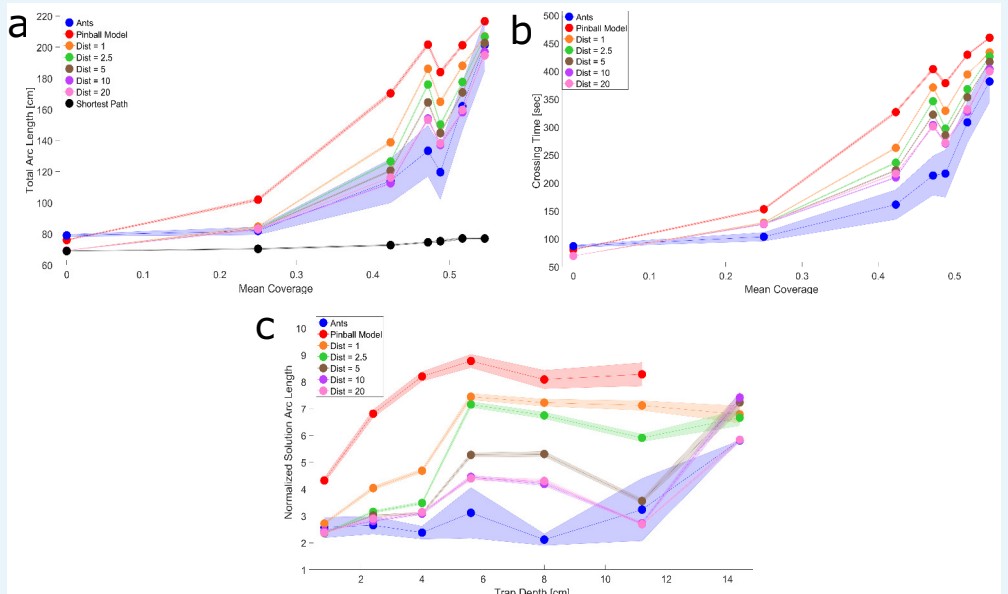

**Appendix 2—figure 4.** Effect of varying the spatially extended sensing parameter in the altered bias simulations. Altered bias simulations (low persistent noise) with different spatially extended sensing parameter ($r_{sense}$) used in the algorithm determining the temporarily altered bias direction at every potential point the simulation might get stuck. Plots show total arc length (**a**) and maze solution time (**b**) vs. mean coverage for different $r_{sense}$ values for the extended pinball model, as well as the performance of the ants (blue), the original pinball model (red) and the shortest path (black) for comparison. The performance of these different simulations when encountering single traps, measured through arc length, as a function of trap depth is plotted in (**c**). Low value $r_{sense}$ simulations do not perform as well as $r_{sense}^{ants} = 10$, and large $r_{sense}$ simulations do not perform better. Shaded regions correspond to standard errors of the mean. Wherever no error is visible, the error is small enough to fit within the filled circle marker.

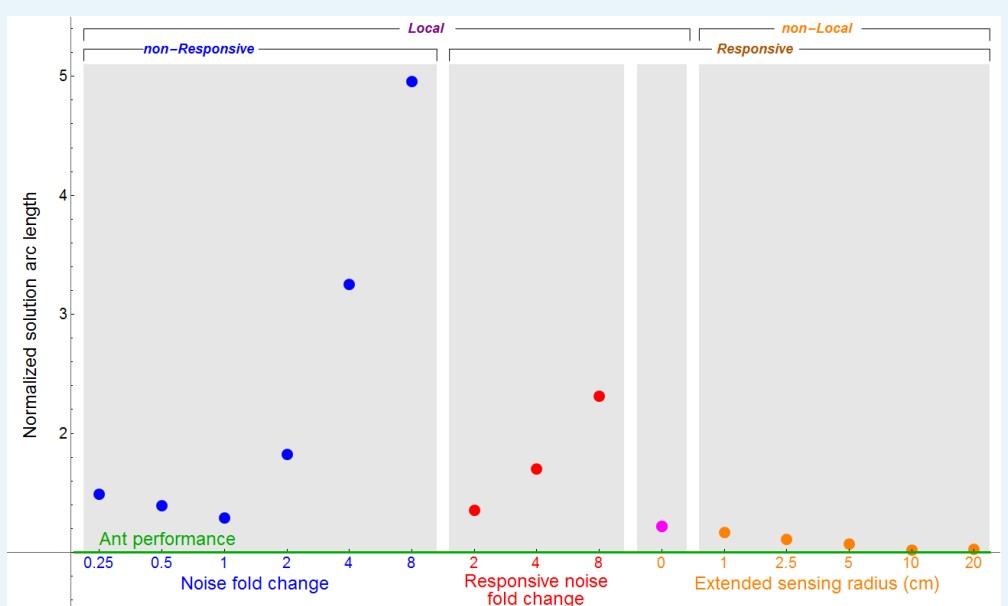

**Appendix 2—figure 5.** Single measure simulation comparison - full results. A full version of *Figure 2d* containing three omitted points with strongly inferior performance. Here we use a single inverse measure for the performance of the simulations, $\frac{L_{\text{sim}}}{L_{\text{ants}}}$, where $L$ is the average solution arc length across all cube densities.

*Appendix 2—figure 4c* uncovers the origin of the discrepancy in overall performance of different sensing range simulations. While all simulations are able to easily bypass shallow traps, the performance of large sensing range simulations (and of the ants) is significantly better when encountering deep traps. Only the latter simulations can keep up with the ants' performance, suggesting the ants do use a form of extended sensing mechanism.

## Appendix 3

## Theory

### 3.1 Theoretical proof for the efficiency of logarithmic vision

Consider the two-dimensional infinite grid $G$. In what follows fix $p>p_c$, where $p_c = 1/2$ is the percolation threshold, as established by Kesten in his seminal work (**Kesten, 1980**). Assume that each edge is open with probability $p$, and closed otherwise. By the properties of the phase transition, with probability 1, the set of open edges in our percolated grid induces a unique infinite cluster, termed $C^\infty$. Moreover, since $1-p<1/2 = p_c$ then, with probability 1, all clusters induced by the set of closed edges are finite. In what follows, we condition on these two highly likely events.

For convenience, we adopt the $\|\cdot\|_\infty$ metric, that is, $\|(x,y)\| = \max\{|x|, |y|\}$. Consider two nodes $s$ and $t$ at 'aerial distance' $d$ from each other on the grid, that is, $\|s - t\| = d$, which are connected over the infinite component $C^\infty$. Angel et al. showed in **Angel et al., 2008** that an agent with locality that is constant in expectation can reach from $s$ to $t$ in $O(d)$ time. The constant hiding behind the "$O$" term may however be large. Here, we wish to show that locality that is logarithmic in $d$ suffices to approximate the shortest path possible to very high precision.

Assume, for simplicity, that $s$ is at $(0,0)$, and that $t$ is at $(d,0)$. Let $D$ be the distance between $s$ and $t$ on the infinite component, that is, the length of the shortest path connecting them in $C^\infty$. We would like to investigate the ability of an agent with limited view to travel from $s$ to $t$ in time that is as close as possible to $D$.

Formally, given a real number $r>0$ and a node $u$ on the grid, define the ball $B_r(u)$ as the subgraph of the percolated grid induced by the set of nodes $\{v \mid \|v - u\| \leq r\}$. We say that an agent has *vision-radius* of $r$ if whenever it resides at a node $u$, the agent 'sees' all edges in $B_r(u)$, and can process this information. We do not restrict the internal computational power of the agent, which in particular means, that when at a node $u$, the agent can performs arbitrary computations on $B_r(u)$, including finding the shortest path in $B_r(u)$ (if it exists) that connects $u$ to another designated node in $B_r(u)$.

Our claims rely on the construction of a strip of logarithmic width (see **Appendix 3—figure 1**). Specifically, we define the *strip*

$$S_\alpha = [0,d] \times [-W/2, W/2]$$

of width $W = \alpha \log d$, for a sufficiently large constant $\alpha>0$. Where $\alpha$ is clear from the context, we may remove the subscript. Note that the strip contains both $s = (0,0)$ as the center node of its left border, termed $L$, and $t = (0,d)$ as the center node of its right border, termed $R$. Let $S^\infty$ denote the intersection between the strip and the infinite component, that is, $S_\alpha^\infty = S_\alpha \cap C^\infty$. For simplicity, we refer to $S_\alpha^\infty$ as the *percolated strip*, although it should be clear that it does not contain all open edges in the strip but only those that belong to the infinite component.

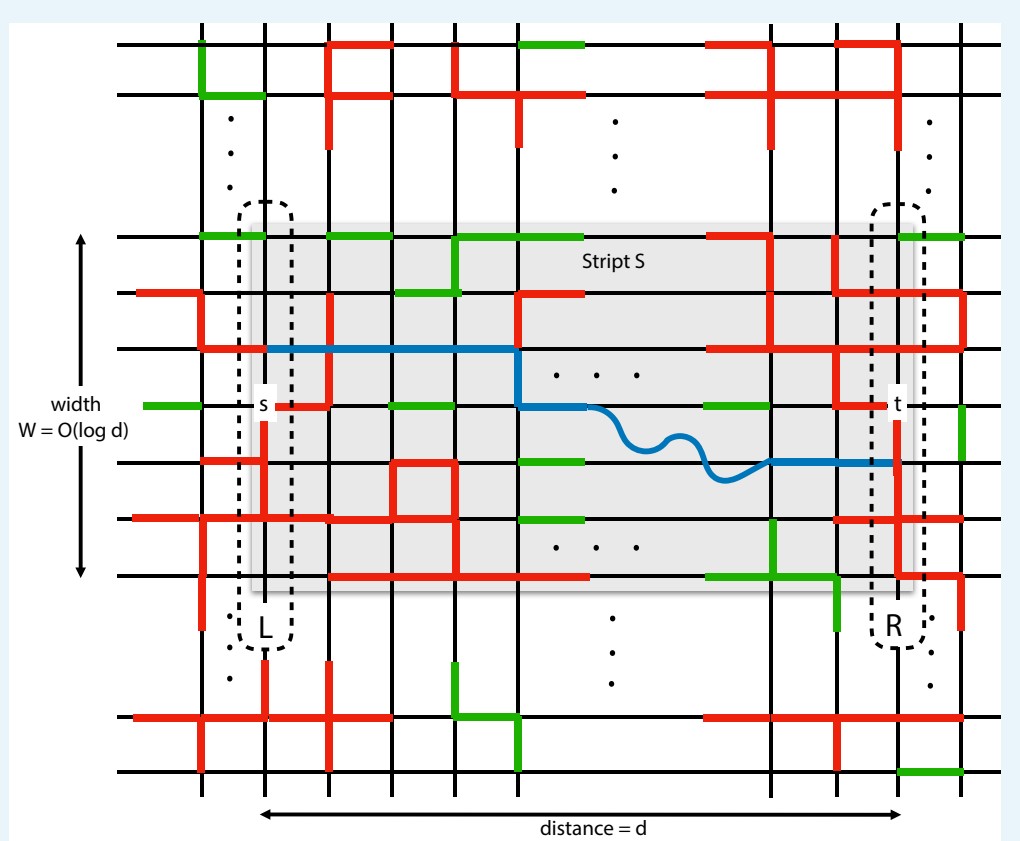

**Appendix 3—figure 1.** The strip $S$ is colored gray. $L$ and $R$ are the left and right borders of the strip, respectively. Green are edges of the percolated network that are not part of the infinite component $C^\infty$. The remaining colored edges (red or blue) are the edges of $C^\infty$. The blue path is the shortest path connecting a node in $L$ to a node in $R$, among those that are fully contained in $S$. The length of this path is $\tilde{D}$. As we shall see, all its edges belong to $C^\infty$ with high probability. The red edges that are the remaining edges of $C^\infty$. The percolated strip $S^\infty$ contains the edges in the strip $S$ that are also in the infinite component. Three dots designate that the network expands in the corresponding direction.

Our arguments are based on three claims: First, when the constant $\alpha$ is sufficiently large then, w.h.p. (We use the term *with high probability (w.h.p)* to denote a probability that is higher than $1 - 1/d^2$. We note that the exponent 2 is arbitrary, and in fact, in all our claims, whenever this guarantee is established, a similar guarantee $1 - 1/d^j$ could have been established, for any $j$, but increasing the constants involved), there exist paths that traverse the entire strip from left to right without ever leaving the strip (Lemma 1). In other words, these paths are contained in $S^\infty_\alpha$ and connect a node on $L$ to a node on $R$. We denote the length of the shortest such path by $\tilde{D}_\alpha$. Although the strip is restricted in the $y$-direction, in the $x$-direction it stretches all the way from $s$ to $t$. We thus refer to $\tilde{D}_\alpha$ as a semi-global minimal traversal solution. Finding a path whose length approximates $\tilde{D}_\alpha$ may not be a trivial task for an agent with a small vision-radius.

Second is our main claim which is formally presented in Theorem 3. It states that for sufficiently large $d$, given $\alpha$, and any $\epsilon > 0$, there exists another constant $\gamma$ such that, w.h.p., an agent with a vision-radius of $r = \gamma \log d$ can travel from $s$ to $t$ along a path whose length is at most $(1 + \epsilon)\tilde{D}_\alpha$. We use simulations to corroborate the applicability of these results for finite size grids.

Finally, to enhance the significance of the latter theoretical result, we use simulations that show that, for not too large values of $\alpha$, $\tilde{D}_\alpha$, the semi-global minimal traversal length, is very close to $D$, the shortest possible traversal length.

Taken together, these arguments show that for percolation mazes above the percolation threshold, a logarithmic field of view suffices for locating a crossing route whose length is very close to what is optimally possible with a complete global view of the maze.

## Lemma 1

There exists a constant $\alpha'$ such that for any $\alpha > \alpha'$ there exists, w.h.p, a simple path connecting the left border of the strip, $L$, to its right border, $R$, that is fully contained in the percolated strip $S_\alpha^\infty$. In particular, $\tilde{D}_\alpha < \infty$.

Proof. We first adopt the notion of a *dual grid*, which is a highly useful tool in the theory of percolation, see, for example, **Grimmett, 2013**; **Steif, 2011**; **Kesten, 1982**. The dual grid is also an infinite grid whose set of vertices is the set of *regions* of the original grid, that is, the squares that are bound by 4 adjacent nodes. There is an edge between two regions if they are adjacent, that is, if they share a grid edge. Another way of viewing the dual graph is simply as a translation of the original grid by the vector $(\frac{1}{2}, \frac{1}{2})$. See **Appendix 3—figure 2**. One then sees that there is an obvious one to one correspondence between the edges of the original grid and those of the dual grid. Given a realization of open and closed edges of the original grid, we obtain a similar realization for the edges of the dual grid by simply calling an edge in the latter graph open if and only if the edge that it crosses in the former graph is open.

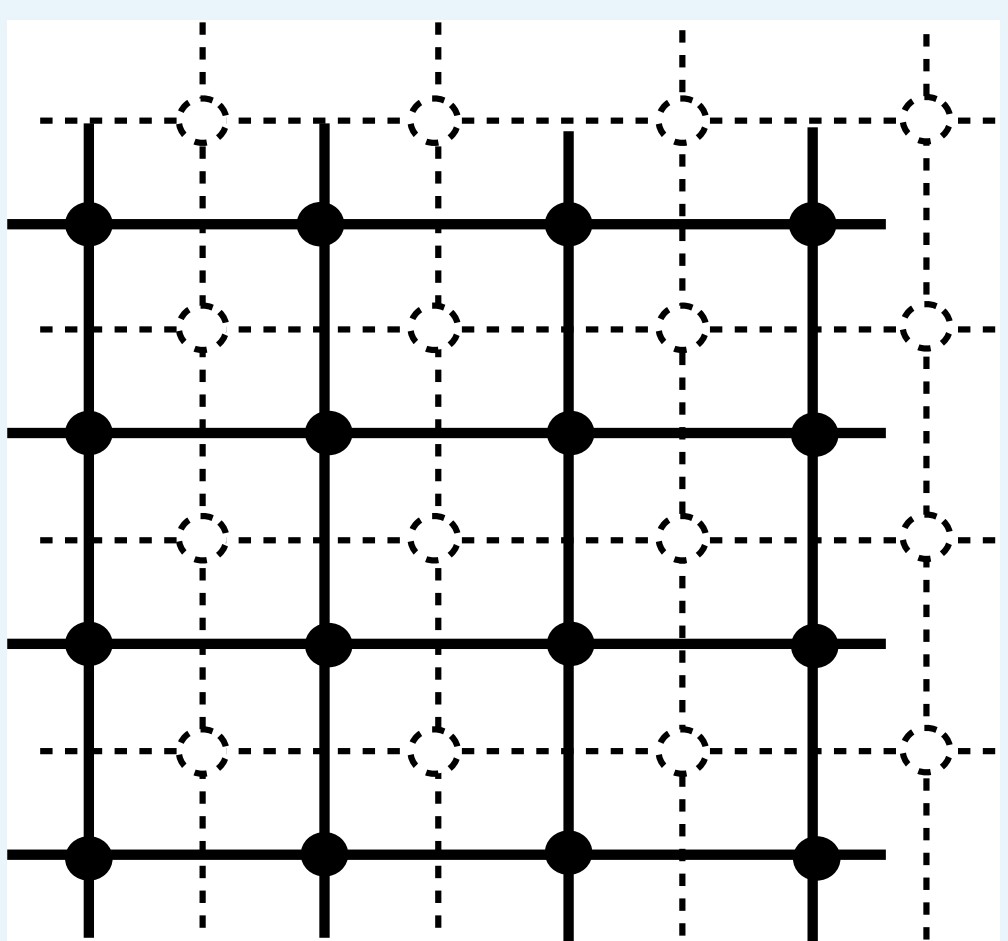

**Appendix 3—figure 2.** The original grid (continuous lines) and the dual graph (dashed lines).

It follows by a version of Whitney's lemma (see also Lemma 7.1 in **Steif, 2011**) that there is a crossing of open edges from the left border $L$ to the right border $R$ of the strip $S_\alpha$ iff there is no simple path of closed edges in the dual graph that connects the upper and lower borders of $S_\alpha$. Such a path of closed edges in the dual graph must be a part of a connected

component of the dual graph whose size is at least the width of the strip, that is, $W = \alpha \log d$. We next argue that for sufficiently large $\alpha$, such a path does not exist w.h.p.

Importantly, the distribution of closed edges in the dual graph follows the same distribution as in the original grid, and is hence, governed by $1 - p < 1/2$. In particular, with probability 1, all connected components of closed edges in the dual graph are of finite size (**Kesten, 1980**). Moreover, the expected size of the cluster of closed edges containing a given node is finite (**Steif, 2011**). For this case, it has been proven by Aizenman and Newman (Proposition 5.1 in **Aizenman and Newman, 1984**) that cluster sizes follow a distribution with exponential tail. In our terminology, their result can be phrased as follows:

### Lemma 2 (Follows from **Aizenman and Newman, 1984**)

Consider the closed edges in the dual graph. There exists a constant $c > 0$ such that the size of the connected component $C$ that contains a given node $u$ satisfies:

$$\Pr(|C| > n) < e^{-cn}.$$

Taking $\alpha > 3/c$ ensures, by Lemma 2, that for every $d \geq 2$, the probability that a given cluster is of size larger than $W = \alpha \log d$ is at most:

$$\Pr(|C| > W) < e^{-c\alpha \log d} < e^{-3 \log d} = \frac{1}{d^3}.$$

The probability that there exists a path connecting a given node $u$ at the upper border of the strip to a node in the lower border is thus at most $\frac{1}{d^3}$. Using a union bound, as there are $d$ nodes on the upper border of the strip, the probability that there is a path of closed edges in the dual graph that crosses the upper and bottom borders of $S_\alpha$ is at most $\frac{1}{d^2}$. Hence, the probability that the original grid contains a continuous path of open edges that connects $L$ to $R$ without ever leaving the strip is at least $1 - \frac{1}{d^2}$.

It remains to show that this $L - R$ crossing belongs to the infinite component $C^\infty$. Note that by definition, this crossing belongs to some component $C$, and that its size is at least $d$. The result of **Aizenman and Newman, 1984**, that is, Lemma 2, cannot be applied here since the expected size of a cluster of open edges is not finite. However, a result by **Kesten, 1982** (see also Equation 1.13 in **Chayes et al., 1987**) states that if the cluster $C$ is finite, then the probability that its size is larger than $d$ is at most $e^{-c\sqrt{d}}$ for some constant $c > 0$. In particular, we get that, w.h.p., $C$ is the infinite component $C^\infty$. This completes the proof of Lemma 1.

We next show that, w.h.p., an agent with logarithmic vision-radius can find a path from $s$ to $t$, whose length almost exactly matches $\tilde{D}_\alpha$.

### Theorem 3

Consider the percolated strip with a sufficiently large $\alpha$ as given by Lemma 1. Assume that $d$ is sufficiently large. For any $\epsilon > 0$, there exists a constant $\gamma > \alpha$ such that w.h.p, an agent with vision-radius of $\gamma \log d$ can find a path from $s$ to $t$ whose length is at most $(1 + \epsilon)\tilde{D}$.

Proof. We shall fix constants $\gamma \gg \beta \gg \alpha$ and define the following algorithm $\mathcal{A}_\gamma$ that relies on a vision-radius of $r = \gamma \log d$. Algorithm $\mathcal{A}_\gamma$ proceeds in phases. In each phase it reduces the distance to $t$ by roughly $\beta \log d$, except for the last phase in which the distance is reduced to zero. At each phase, the agent starts at some node $u \in S_\alpha^\infty$ and concludes at another node $v \in S_\alpha^\infty$, whose x-axis coordinate is $\beta \log d$ higher than that of $u$ (except for the last phase, where the agent terminates on $t$).

In order to describe a phase we need a few definitions. Recall that $B_r(u)$ denotes the connected component of $u$ induced by the nodes up to distance $r$ from $u$. Given $u \in S_\alpha^\infty$, define the *goal set* $\mathcal{G}_\beta(u)$ as the set of nodes at distance $\beta \log d$ to the right of $u$, i.e., in the direction towards $t$, that belong to $S_\alpha^\infty$. Note that the x-axis value of these nodes is $\beta \log d$ over the x coordinate of $u$. If $u$ itself is of distance less than $\beta \log d$ from $t$, then $\mathcal{G}_\beta(u)$ is simply $\{t\}$. Each phase is described as follows (see **Appendix 3—figure 3** for an illustration).

- Algorithm $\mathcal{A}_\gamma$. Standing at a node $u \in S_\alpha^\infty$ the agent walks along the shortest path in the ball $B_r(u)$ from $u$ towards any of the nodes in $\mathcal{G}_\beta(u)$. (If there is no such path the algorithm halts.)

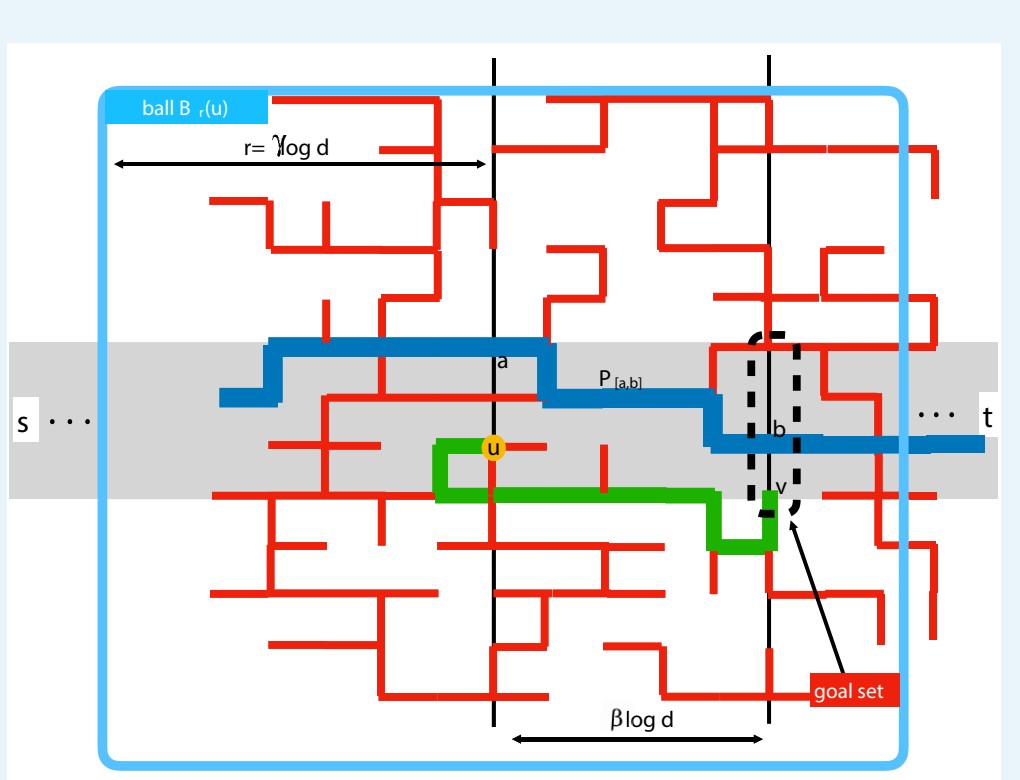

**Appendix 3—figure 3.** Description of a phase in Algorithm $\mathcal{A}_\gamma$. The colored short lines are the open edges of the infinite cluster $C^\infty$. The strip $S$ is colored gray. The blue path is $\tilde{P}$ - a shortest path from $L$ to $R$ among the ones fully contained in the strip $S_\alpha^\infty$. The agent starts the phase at node $u$ (yellow circle) and finds a shortest path (colored green) in $B_r(u)$, its ball of view of radius $r = \gamma \log d$, towards a node $y$ in the goal set $\mathcal{G}_\beta(u)$. Note that this path is not necessarily fully contained in the percolated strip $S_\alpha^\infty$. The red edges that are the remaining edges of $C^\infty$.

Note that if $t \in B_r(u)$ then $\mathcal{G}_\beta(u) = \{t\}$, and hence, in this case, the agent simply walks along the shortest path in the ball towards $t$. Observe also that the agent is not restricted to walk always inside the strip, although at the beginning and ending of a phase it always resides inside.

We next analyze the performances of Algorithm $\mathcal{A}_\gamma$. Before we begin the analysis, recall that we consider the percolated strip with sufficiently large $\alpha$, hence Lemma 1 promises that w.h.p, there exists a simple path connecting the left border $L$ and the right border $R$ that is fully contained in the percolated strip $S_\alpha^\infty$. Let us condition on this high probability event.

The algorithm executes at most $d/\beta \log d$ phases. Let us consider a given phase where the agent starts at a node $u \in S_\alpha^\infty$. Let $\tilde{P}$ be a shortest path among the paths connecting $L$ and $R$ that are fully contained in $S_\alpha^\infty$. By definition, the length of $\tilde{P}$ is $|\tilde{P}| = \tilde{D}_\alpha$. Let $a$ be the node on the path $\tilde{P}$ with the same $x$-coordinate as $u$, and let be $b$ the node on $\tilde{P}$ that belongs to the goal set $\mathcal{G}_\beta(u)$. Let $\tilde{P}_{[a,b]}$ be the segment of the path $\tilde{P}$ that goes from $a$ to $b$.

## Lemma 4

For sufficiently large $\beta > \alpha$, with probability at least $1 - \frac{1}{d^5}$, the agent does not halt in the phase, and terminates at a node in the goal set $\mathcal{G}_\beta(u)$. Moreover, the length of the path taken by the agent in the phase is at most:

$$(1+\epsilon)|\tilde{P}_{[a,b]}|.$$

Before proving the lemma, let us see how it can be used to conclude the proof of the desired Theorem 3. The path $\tilde{P}$ can be broken into segments $\tilde{P}_{[a_i, b_i]}$, defined by the phases $i = 1, 2, \cdots$ of the algorithm. Specifically, let $L_i$ be the set of nodes on the percolated strip whose $x$-axis equal that of $u_i$ - the node where the agent is at the beginning on phase $i$. Let $R_i$ be the nodes in the percolated strip whose $x$-axis equal that of $u_i$ plus $\beta \log d$, that is, $R_i$ is simply the corresponding goal set. Then $\tilde{P}_{[a_i, b_i]}$ is defined as the part of the path $\tilde{P}$ from the first time it enters $L_i$ (at node $a_i$) until the first time it hits $R_i$ (at node $b_i$). For each such segment, Lemma 4 implies that the algorithm uses a path whose length approximates the length of $\tilde{P}_{[a_i, b_i]}$ to within a multiplicative factor of $1 + \epsilon$, with probability at least $1 - \frac{1}{d^3}$. Hence, as there are at most $d$ segments, using a union bound argument, the combined path produced by the algorithm approximates $|\tilde{P}|$ to within a multiplicative factor of $1 + \epsilon$, with probability at least $1 - \frac{1}{d^2}$. This establishes Theorem 3.

Proof of Lemma 4. Before starting the proof let us first discuss the connection between the 'aerial distance' $\|u - v\|$ between two nodes $u$ and $v$ in the same cluster and $D(u, v)$, the distance between them on the cluster. A classical result by **Antal and Pisztora, 1996** states that above the percolation threshold, the distance on the percolation graph between two nodes in the same cluster is linear in their 'aerial distance'. Specifically, Theorem 1.1 in **Antal and Pisztora, 1996** states:

## Theorem 5 (Antal and Pisztora)

Let $p > p_c$. Then there exists a constant $c$ (which depends on $p$) such that, conditioning on $u$ and $v$ being in the same cluster, we have:

$$\limsup_{\|u-v\| \to \infty} \frac{1}{\|u - v\|} \log Pr(D(u,v) > c\|u - v\|) < 0.$$

As the lim sup exists and is negative, Theorem 5 implies that there exists an integer $M$ and a constant $\delta > 0$ such that for all $u$ and $v$ with $\|u - v\| > M$,

$$\log Pr(D(u,v) > c\|u - v\|) < -\delta\|u - v\|,$$

implying the following corollary.

## Corollary 6

There exist constants $\delta, c > 0$ and $M$ such that for all $u$ and $v$ with $\|u - v\| > M$,

$$Pr(D(u,v) > c\|u - v\|) < e^{-\delta\|u-v\|}.$$

We next show that by taking $\gamma$ to be a sufficiently large constant, we can expect that the ball $B_{r/2}(u)$ will include a path from $u$ to $a$. For this purpose we apply Corollary 6 on the nodes $u$ and $a$. Note that these nodes share the same $x$-axis coordinate and belong to the percolated strip $S_\alpha^\infty$. Therefore, the 'aerial distance' between them is at most $\alpha \log d$. Note that by definition, they both belong to the infinite cluster, hence $D(u, a)$ denotes the distance between them on that cluster. Applying the corollary therefore implies that there exist constants $\delta, c > 0$ and $M$, such that for all $d > M$, we have

$$Pr(D(u,a) > c\alpha \log d) < e^{-\delta\alpha \log d}. \tag{1}$$

Taking $\alpha > 3/\delta$ and $\gamma > 2c\alpha$ thus ensures that:

$$Pr(D(u,a) \leq \gamma/2 \log d) > 1 - \frac{1}{d^3}.$$

Therefore, w.h.p, a shortest path from $u$ to $a$ on the infinite cluster $P_{[u,a]}$ belongs to the ball $B_{r/2}(u)$. Note that even though both end-points $u$ and $a$ belong to the strip, the shortest path connecting them may go out of the strip. However, it is still guaranteed, w.h.p., to belong to the ball $B_{r/2}(u)$.

A similar argument shows that by choosing $\gamma > 2c\beta$, the path $\tilde{P}_{[a,b]}$, that is the subpath of $\tilde{P}$ that goes from $a$ to $b$, is w.h.p included in the ball $B_{r/2}(a) \subset B_r(u)$. The concatenated path

$$P_{[u,b]} := P_{[u,a]} \cup \tilde{P}_{[a,b]}$$

is thus included in the ball $B_r(u)$. Again, this concatenated path may go out of the strip, but remains in the ball $B_r(u)$ w.h.p. When this happens the set of paths in $B_r(u)$ that connect $u$ to a node in the goal set is not empty, and hence, w.h.p, the agent does not halt.

Next, let us analyze the length of the path taken by the agent in the phase. As the agent takes the shortest path in the ball $B_r(u)$ towards a node in the goal set, the length of this path is at most the length of $P_{[u,b]}$, which is by the triangle inequality, at most:

$$|P_{[u,b]}| \leq |P_{[u,a]}| + |\tilde{P}_{[a,b]}|.$$

By *Equation 1*, this is, w.h.p., at most:

$$|P_{[u,b]}| \leq c\alpha \log d + |\tilde{P}_{[a,b]}|.$$

Taking $\beta > c\alpha/\epsilon$, therefore implies that, w.h.p.:

$$\frac{|P_{[u,b]}|}{|\tilde{P}_{[a,b]}|} \leq \frac{c\alpha \log d}{\beta \log d} + 1 \leq 1 + \epsilon.$$

Or in other words, the length of the selected path is at most $(1+\epsilon)|\tilde{P}_{[a,b]}|$, as desired. This concludes the proof of Lemma 4, and thus completes the proof of Theorem 3.

## 3.2 Simulation showcasing the efficiency of logarithmic vision

The above proof, Appendix 3.1, provides a theoretical basis to the idea that logarithmic vision is enough for efficient crossing of a percolation lattice above the percolation threshold. However, the parameters used can be of any size; for example in Theorem 3, for a certain $\epsilon$, $\gamma$ can be such that it encompasses the entire system. Thus, we wanted to further corroborate the feasibility of the algorithm by implementing it programmatically and compare to biased random walk as well as to the shortest path on the strip $\tilde{D}_\alpha$ and overall $D$. See Materials and methods for implementation description and parameter fitting.

We chose $\gamma$, the vision radius parameter, to be equal to $\alpha$ (=20), the width of the strip. Namely, we simulated the weakest version of our algorithm. In this scenario, the field of view is 0.45% of the length of the grid. The results show that the logarithmic vision algorithm can find a path that crosses the grid efficiently; the mean percent increase in path length when comparing the vision algorithm path to the strip shortest path is ~2.44% (*Appendix 3—figure 4a*).

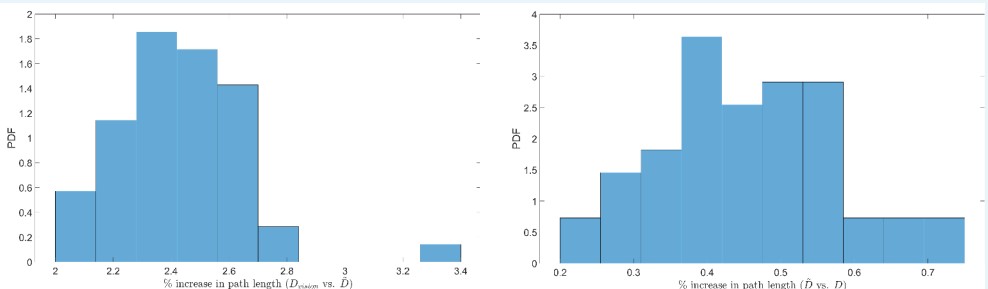

**Appendix 3—figure 4.** Comparison of simulation path lengths. PDF histograms of percentage increase in path length comparing (**a**) logarithmic vision algorithm path to strip-constrained shortest path and (**b**) strip-constrained shortest path to overall shortest path.

To give greater significance to this result, we wanted to compare the strip shortest path $\tilde{D}_\alpha$ with the overall shortest path $D$, to show that for a reasonable (i.e. not too large) values of $\alpha$,

$\tilde{D}_\alpha$ approximates $D$. Indeed, we get that for the chosen value of $\alpha$ (=20), which translates into a strip width of ~0.45% of the length of the grid, the average percent of increase to the length of the shortest path when constrained to the aforementioned strip is merely ~0.46% (**Appendix 3—figure 4b**).

The logarithmic vision algorithm was compared to a baseline Ant-in-a-labyrinth biased random walk simulation. Again, BRW simulation description and parameter fitting are detailed in Materials and methods. The BRW simulations failed miserably when compared to the vision algorithm. None of the 10,000 total iterations were able to solve the maze in the allotted time. Thus, as can be observed in **Figure 4b** we compared the speed of the simulations, taking the mean maximum advancement in x divided by the duration of the simulation for the BRW simulation $\left(\frac{<x_{max}>}{T_{max}}\right)$ and the lattice length divided by the path length for the logarithmic vision algorithm and the shortest path calculations $\left(\frac{d}{D}\right)$. This comparison highlights the superiority of the logarithmic vision algorithm over biased random walk.

