## [Decision Letter]

**Acceptance summary:**

This article presents a compelling example and analysis of how collective intelligence can lead to the accomplishment of nontrivial tasks. Specifically, the authors show that the method that crazy ants use to move food across complex landscapes involves a sensing range that extends to the group rather than just the individual. As foraging is often described as a random walk, the authors develop several models and show that one that invokes an extended sensing range successfully recapitulates the search paths found in the animals. They also present a theoretical analysis of percolation theory, showing that sensing should extend up to the logarithm of the system size.

**Decision letter after peer review:**

Thank you for submitting your article "Ant collective cognition allows for efficient navigation through disordered environments" for consideration by *eLife*. Your article has been reviewed by three peer reviewers, including Gordon J Berman as the Reviewing Editor and Reviewer #1, and the evaluation has been overseen by Christian Rutz as the Senior Editor. The following individual involved in the review of your submission has agreed to reveal their identity: Adam J Calhoun (Reviewer #3).

The reviewers have discussed the reviews with one another and the Reviewing Editor has drafted this decision to help you prepare a revised submission.

Summary:

In this manuscript, Aviram et al. examine a collective navigation task where longhorn crazy ants are required to transport a large object through a maze-like environment. The task requires cooperation between ants, both to transport the object as well as to find an efficient path out of traps. The authors show that: (1) ants are able to successfully complete the task, even for dense mazes close to the no-solution limit; (2) a simulated biased random walk does not solve the problem as it gets trapped, and argue that the solution requires a non-local sensing component; (3) this non-local sensing is performed by leader ants, which sense an extended area and lead the ants out of traps; (4) a biased random walk with this additional component performs similarly to the ants in simulations; and (5) theoretical arguments which reveal that an extended sensing radius that scales logarithmically with maze size is sufficient to perform this task.

Although the reviewers agreed that the work was of sufficient quality to merit consideration, several major points need to be addressed – enumerated below.

Essential revisions:

1) There was much confusion about the choice of the 10 cm sensing radius. First, Figure 2D shows that "non-local, responsive" algorithms with an extended sensing radius of 20 cm are a factor 10 worse than the ants' performance, which directly contradicts Figure 2C's plot for the extended pinball model, which shows similar performance to the ants. The authors should clarify why this is the case. Moreover, it's unclear what should be taken from Figure 2D, particularly since the simulations discussed in that figure are not discussed until a later section. Second, the simulation results depend strongly on extended sensing radius *r_sense_*, which is taken to be 10 cm. The 10 cm is based on the results on ants in traps shown in Figure 3. Importantly, from Figure 3A, it is unclear how *r_sense_* is 10 cm, while the scale bar of 5 cm clearly indicates a *r_sense_* less than 5 cm. Moreover, the Log(N) label for the heatmap is ill-defined (what is N? what is the base of the Log?)

Is 10 cm supposed to be optimal? Is the prediction that the sensing range has to do with the size of the system found in the natural world, or that it changes with the size of the system that is presented experimentally?

2) The theoretical work is interesting but seems disconnected from the rest of the paper. A connection between the assumptions in the model and the experimental setup is not clearly made. A ballpark estimate from the theory with experimental numbers is also not presented. In general, the reviewers felt that more scaffolding material is needed to tie-in the theory with the experiments/simulations. Specifically, the paper could benefit from having an expanded discussion of the theory in the main text. The general gist comes across, but the reviewers didn't really have a sense of what was going on until they read the supplemental section.

3) On a similar note, it would have been nice to see a slightly longer summary/Discussion section putting the work into context. The reviewers thought that the ideas proposed are strong, but would benefit from a more thorough explanation. In particular, the reviewers felt that more connections to the biological literature were necessary, pointing-out the biological implications of the findings in a more thorough manner.

4) On the whole, many of the figures are difficult to read – the labels are small and in many of the supplementary figures, the legends are impossible to see. The error bars are barely visible. The color schemes are also confusing, for instance, in Appendix 2—figure 2 where both blue and turquoise are used. We ask that the authors extensively modify the figures for clearer presentation.

---

## [Author Response]

Essential revisions:1) There was much confusion about the choice of the 10 cm sensing radius. First, Figure 2D shows that "non-local, responsive" algorithms with an extended sensing radius of 20 cm are a factor 10 worse than the ants' performance, which directly contradicts Figure 2C's plot for the extended pinball model, which shows similar performance to the ants. The authors should clarify why this is the case.

The comparison measure we chose to use in Figure 2D was indeed confusing, and we thank the reviewers for pointing this out. The y-axis of Figure 2D was initially defined as:

(simulation solution length-minimal solution distance) / (ant solution length – minimal solution length).

Together with our usage of a logarithmically scaled y-axis, this presentation greatly accentuates small differences and, as we now understand, is confusing. We have now changed the y-axis of Figure 2D to depict a simpler, more intuitive normalization which we present on a linear scale:

(simulation solution length – ant solution length) / (ant solution length).

This is simply the percentage by which different solutions are inferior to the ants’ solution. It can now be seen more clearly that extended sensing solutions with a sensing ranges of 10cm and 20cm approach the ant solution. They are, nevertheless, slightly less efficient than the ants themselves (this is discussed in the paragraph before last of the subsection titled “Extended sensing facilitates efficient trap and labyrinth traversal”).

We have updated the caption of this figure to more clearly state what the y-axis signifies, we also removed three simulation points with very inferior performance to allow us to present the results on a linear scale (rather than logarithmic) which again adds to the clarity of the point we wish to make in this figure. The full results of all performed simulations are now included as an appendix figure (Appendix 2—figure 5) that is referred to from the caption of Figure 2D.

Moreover, it's unclear what should be taken from Figure 2D, particularly since the simulations discussed in that figure are not discussed until a later section.

Figure 2D is central to this paper since it provides an overview of the results and the progression of the manuscript. It shows how the performances of different physics-based simulations compare to the performance of the ants. It clearly shows that local, non-responsive algorithms which are completely physical, like a ball falling through an array of pegs (as discussed in the second Results subsection), do far worse than the ants. The figure also demonstrates that local algorithms which are responsive to traps (discussed in the third Results subsection) still do not suffice to capture ant performance; importantly, only when non-local properties (as discussed in the fourth Results subsection) are taken into account can the simulation performances approach those of the ants. Figure 2D even shows that the effective length-scale for this non-local extension in sensing is about 10cm.

The figure’s location in the text coincides with our initial foray into quantitative comparisons between ant and simulation performances. Despite the fact that the figure also contains simulation results which are discussed later on, we believe this is the right place for this summarized depiction of all simulation performances. The mere phenomenological result evident in this figure, that the ants outperform multiple variations of physics-based models, is a main motivation for the deeper analysis of the ants’ collective behavior into which we delve in the following sections of the article. We feel that a good location for a motivating figure is near the beginning of the argument. Moreover, this motivation leads us to search which characteristics of the ants’ collective behavior are responsible for their improved performance and Figure 2D summarizes our search and its results. We feel that presenting all these results in a single panel allows for comparisons between empirical and simulation performances in a single succinct view. This alleviates the need for replication of the same data every time we present a new algorithm and simulation.

We have now explicitly stated the role of Figure 2D where we first refer to it. We have further expanded the caption for this figure so it is easier to understand. Also note that when we first refer the reader to Figure 2D we direct attention specifically to the relevant simulations as marked by the blue ellipse – which includes only simulations which were introduced by that point in the paper.

Second, the simulation results depend strongly on extended sensing radius r_sense_, which is taken to be 10 cm. The 10 cm is based on the results on ants in traps shown in Figure 3. Importantly, from Figure 3A, it is unclear how r_sense_ is 10 cm, while the scale bar of 5 cm clearly indicates a r_sense_ less than 5 cm.

Importantly, during cooperative transport by longhorn crazy ants single ants can lead the entire group (Gelblum et al., 2015). Therefore, we must turn our attention to the tails of the ant counts distribution presented in Figure 3; ants which arrive from these far-off areas can act as leaders and they set the limit on the effective sensing range of the group. We now clarify this point in the last paragraph of the subsection titled “Collective extension of sensing range”.

We have now added a Figure 3—figure supplement 2 which shows the cumulative distribution of ant number as a function of the radius from the load’s center. This figure shows that the distribution levels off on the 10cm scale – The distribution reaches 99% at a distance of ~14cm, which is of the order of 10cm. We therefore took 10cm as the relevant length-scale for ant extended sensing. Indeed, as Figure 2D and Figure 3C-D show, extended sensing at this length-scale could help us explain the ants’ improved performances. To emphasize, we do not claim that 10cm is a very precise value or that the ants have evolutionarily converged on an exploration scale that is precisely optimized for our setup. There is no real reason to assume this. What we do claim is that the observed sensing length-scale which is on the order of 10cm (rather than 1cm or 100cm) can explain the ants’ improved performances. These points are now extensively discussed in the new subsection titled “Relating theoretical results and empirical findings”.

Moreover, the Log(N) label for the heatmap is ill-defined (what is N? what is the base of the Log?)

The heat-map’s logarithmic scale is in base 10. N was taken as the count of ants in the 2D histogram used to create the heat map – we thank the reviewers for noticing that we neglected to state this. To make the color-bar labelling clearer, we changed it to simply read “ant counts” and altered the tick labels on the colorbar to show the logarithmic nature of the scale. We also added a clearer description of the heat map contents in the caption of the figure.

Is 10 cm supposed to be optimal? Is the prediction that the sensing range has to do with the size of the system found in the natural world, or that it changes with the size of the system that is presented experimentally?

As stated two answers above, 10cm is an order of magnitude for the collective sensing and we do not make the claim that 10cm is precisely optimal. That being said, we agree that correlation to natural world statistics is of interest. Since precise environmental statistics are unavailable – our arguments connect three main length scales present in the problem (to an order of magnitude): the ants’ foraging range on the scale of 10 meters, the relevant obstacle size on the scale of centimeters, and the range of extended sensing on the scale of 10 cm.

As the reviewers correctly point here, system size is the important factor. The natural foraging range of this ant species is on the order of 10 meters while our experimental arena is only 70cm long. It is therefore correct to wonder if conclusions regarding the efficiency of a certain sensing range which apply to our experimental setup indeed hold relevance for larger ranges. To understand this we note that the basic unit is the modular building block of the traps: the single cube. The cube edge size is about 1 cm which is the relevant length scale of loads collectively carried by these ants in natural settings (large insects) and therefore, by order of magnitude, the size of the relevant obstacles. We should therefore compare 1000 units (10 meters in units of centimeters) in the natural case with 70 units (70 centimeters in units of centimeters) in our experimental system. As our theoretical results show, optimal sensing ranges scale logarithmically with system size. Consequently, we note that log(1000)/log(70)~1.6, which means that if the sensing range of 10cm is near optimal over 10m then it should also be near optimal over 70cm. In other words, due to its logarithmic dependency, the optimal sensing range does not change significantly between natural and experimental systems. Again, we stress that we do not expect the ants to be precisely optimal and are not looking for accurately tuned sensing ranges. We merely argue that a sensing range on the order of 10cm (and not 1cm or 100cm, for example) makes sense in this system.

This discussion is now included in the newly added subsection titled “Relating theoretical results and empirical findings”.

2) The theoretical work is interesting but seems disconnected from the rest of the paper. A connection between the assumptions in the model and the experimental setup is not clearly made. A ballpark estimate from the theory with experimental numbers is also not presented. In general, the reviewers felt that more scaffolding material is needed to tie-in the theory with the experiments/simulations.

The theory aims to present a simplified discretized view into our experimental observations and explain why a logarithmic sensing range that is larger than strictly local but much smaller than the size of the system can be enough to yield a huge speedup in crossing times.

While our theoretical findings provide the desired qualitative conclusions, we agree with the reviewers that more quantitative results are of value: theoretical results that are in the quantitative ballpark of the empirical findings would strengthen our case. We now make such quantitative comparisons in two ways:

1) We specify the grid spacing which connects length scales in our theory to those in the real world. We now make the point that the grid spacing coincides with the size of one cube. Roughly speaking, the addition of one cube translates into the removal of a single edge from the 2D grid. Hence the grid spacing can be taken to be on the order of the length of a cube’s edge: 1cm. As mentioned above, this means that the sensing range we are dealing with is on the order of both the ants’ natural foraging range log(1000cm/1cm)=log(1000) ~ 10cm and the size of our experimental system log(70cm/1cm)=log(70) ~ 6cm. Both of these are clearly in the ballpark of the value we have originally estimated from the empirical results: 10cm. This discussion is now included in the new subsection titled “Relating theoretical results and empirical findings”.

2) Our theoretical results show that a natural extended sensing algorithm not only performs extremely well, but also does not deviate much from the line that connects the starting point to the destination point. In the language of the theoretical results, this means that, during the motion, the extended sensing algorithm remains contained within a narrow strip around this line. Returning to the ant experiments, we predict that ants would not engage in frequent large departures perpendicular to their principle direction of motion. We now analyze the spread of the load trajectories perpendicular to overall motion and indeed find that it too is confined to a scale that is consistent with the logarithm of system size. The results of this analysis are now mentioned in the first two paragraphs of the new subsection titled “Relating theoretical results and empirical findings” and supported by the new Figure 4—figure supplement 1.

Specifically, the paper could benefit from having an expanded discussion of the theory in the main text. The general gist comes across, but the reviewers didn't really have a sense of what was going on until they read the supplemental section.

We have now added a full-page description outlining in greater detail the main ideas that go into the proof to the main text.

3) On a similar note, it would have been nice to see a slightly longer summary/Discussion section putting the work into context. The reviewers thought that the ideas proposed are strong, but would benefit from a more thorough explanation. In particular, the reviewers felt that more connections to the biological literature were necessary, pointing-out the biological implications of the findings in a more thorough manner.

We have significantly expanded the Discussion. The Discussion includes two parts. The first is the newly added subsection “Relating theoretical results and empirical findings” where we discuss the connections between theory and experiment as well as the generality and near-optimality of the ants’ collective navigation algorithm. The second is the expanded “Summary” section: here we identify the ants’ collective behavior as a form of active, collective remote sensing and discuss the aspects of these traits connecting them to other animal systems. Further, we discuss the applicability of our results to the percolation and ant-in-a-labyrinth theoretical frameworks, and the impact of our results in light of previous results and main directions in these fields. Finally, we claim that the connections to general abstract theories mean we can attempt to learn from our results on other, seemingly unrelated search problems and provide the example of population dynamics during an evolutionary process.

4) On the whole, many of the figures are difficult to read – the labels are small and in many of the supplementary figures, the legends are impossible to see. The error bars are barely visible. The color schemes are also confusing, for instance, in Appendix 2—figure 2 where both blue and turquoise are used. We ask that the authors extensively modify the figures for clearer presentation.

We edited the supplementary figures and enlarged them to make all labels and legends readable. Error bars were often hidden because the errors were very small and consequently fitted within the size of the points themselves. In those cases, a note is now added to the figure caption to prevent confusion.

Regarding color choices, the colors were chosen from a maximally discernible set and we believe that the reason it was hard for the reviewers to distinguish between the lines is the small size of the image, which is now fixed where needed.